# A native conjugative plasmid confers potential selective advantages to plant growth-promoting *Bacillus velezensis* strain GH1-13

Yunhee Choi[1,4], Ha Pham[1,4], Mai Phuong Nguyen[1,4], Le Viet Ha Tran[1], Jueun Kim[2], Songhwa Kim[3], Chul Won Lee [2✉], Jaekyeong Song [3✉] & Yong-Hak Kim [1✉]

The conjugative plasmid (pBV71) possibly confers a selective advantage to *Bacillus velezensis* strain GH1-13, although a selective marker gene is yet to be identified. Here we show that few non-mucoid wild-type GH1-13 cells are spontaneously converted to mucoid variants with or without the loss of pBV71. Mucoid phenotypes, which contain or lack the plasmid, become sensitive to bacitracin, gramicidin, selenite, and tellurite. Using the differences in antibiotic resistance and phenotype, we isolated a reverse complement (COM) and a transconjugant of strain FZB42 with the native pBV71. Transformed COM and FZB42p cells were similar to the wild-type strain GH1-13 with high antibiotic resistance and slow growth rates on lactose compared to those of mucoid phenotypes. RT-PCR analysis revealed that the expression of plasmid-encoded orphan aspartate phosphatase (pRapD) was coordinated with a new quorum-sensing (QS) cassette of RapF2–PhrF2 present in the chromosome of strain GH1-13, but not in strain FZB42. Multi-omics analysis on wild-type and plasmid-cured cells of strain GH1-13 suggested that the conjugative plasmid expression has a crucial role in induction of early envelope stress response that promotes cell morphogenesis, biofilm formation, catabolite repression, and biosynthesis of extracellular-matrix components and antibiotics for protection of host cell during exponential phase.

[1] Department of Microbiology, Daegu Catholic University School of Medicine, Daegu, Republic of Korea. [2] Department of Chemistry, Chonnam National University, Gwangju, Republic of Korea. [3] Agricultural Microbiology Division, National Institute of Agricultural Sciences, Rural Development Administration, Wanju-gun, Jeollabuk-do, Republic of Korea. [4]These authors contributed equally: Yunhee Choi, Ha Pham, Mai Phuong Nguyen. ✉email: cwlee@chonnam.ac.kr; mgjksong@korea.kr; ykim@cu.ac.kr

Several plant growth-promoting rhizobacteria (PGPR) act as biocontrol and plant growth-promoting agents. Recently, an increasing number of *Bacillus velezensis* strains have been reported to maintain a stable relationship with plant roots, promote plant growth, and suppress soil-borne plant pathogens by producing biofilms[1], lipopeptides[2], quorum-quenching enzymes[3], phytase[4], polyketides[5,6], indole-3-acetic acid (IAA)[7], and 2,3-butanediol[8]. The model strains, *B. subtilis* 168 (*SubtiWiki*)[9] and *B. velezensis* FZB42 (*AmyloWiki*)[10], provide an annotation of genes with PGPR traits. However, the understanding behind promoting these traits is still evolving.

We isolated *B. velezensis* strain GH1-13 from rice paddy soil, and reported its ability to promote plant growth and suppress soil-borne pathogens[11,12]. The *B. velezensis* strain GH1-13 chromosome, similar to that of FZB42 genome[10,13], encodes genes for biosynthesis of plant hormones such as IAA, 2,3-butanediol, and antibiotics such as lipopeptides, and polyketides. The strain GH1-13 harbors a *Bacillus* plasmid, here named pBV71. Heat curing of the plasmid showed a decrease in the production of IAA[11], but the plasmid did not carry this function. Plasmid pBV71 encodes an orphan Rap protein (pRapD), which is a known negative regulator of *Bacillus* competence, biofilm formation, and sporulation[14]. Because the chromosome encodes multiple Rap-Phr systems, it is possible that the plasmid-chromosome crosstalk can modulate cell growth and quorum-sensing (QS) to confer an advantageous trait[15]. A previous study showed that the plasmid-encoded *rap* gene of *B. amyloliquefaciens* subsp. *plantarum* strain S499 plays a role in controlling several traits such as protease secretion, lipopeptide production, and biofilm formation[16]. Therefore, understanding the tight regulatory plasmid-chromosome crosstalk, which may regulate gene expression and cell differentiation is important in the PGPR *Bacillus*.

This study aimed to investigate the impact of pBV71 on the host cell by plasmid curing, complementation, and conjugation. We found that strain GH1-13 with non-mucoid phenotype was often converted to mucoid variants, with or without plasmid loss, during normal growth conditions in rich media. The non-mucoid to mucoid conversion was accompanied by decreased resistance to bacitracin, gramicidin, selenite, and tellurite. These findings were used in the experimental design of conjugation and complementation strategies as a selective marker gene in the large pBV71 plasmid was absent. Further, comparative analyses suggested that pBV71 is involved in crosstalk with the RapF2–PhrF2 QS system present in *B. velezensis* strain GH1-13, but not in strain FZB42, and controls cell morphogenesis, biofilm formation, catabolite repression, and antibiotic production for the host protection during the exponential phase of cell growth.

## Results and discussion

### Plasmid sequence and transcription

The sequence of pBV71 in *B. velezensis* strain GH1-13 encodes many proteins, which are hypothetical proteins in the genus *Bacillus* (Supplementary Data 1). Multiple genome sequence alignment shows a shared region of ~6 kb in pBV71 that is similar to parts of Antarctic *B. safensis* strain U14-5 unnamed plasmid and four draft genome sequences of *Bacillus* species (Supplementary Fig. 1). This ~6 kb region of pBV71 harbors putative *tra* and *vir* (*trs*) genes (Supplementary Data 2), presumably encoding the type IV pilus that participates in bacterial conjugation.

Using Illumina sequencing of transcripts from the plasmid, open-reading frames (ORFs) were quantified at different times (5, 8, 12, and 24 h) during the cultivation of strain GH1-13 in a tryptic soy broth (TSB) with aeration at 25 °C. To minimize sample bias resulting from different sampling times, the RNA-seq data were normalized using the Trimmed Mean of *M*-values

(TMM)[17]. The normalized ORF transcript levels at different stages of cell growth were displayed in heat maps on a circular genome map, exhibiting dynamic patterns of expression over time during the host cell cycle (Supplementary Fig. 2). We noticed that several ORFs encoding putative conjugal proteins in a ~18 kb region between the plasmid segregation protein ParM gene and a type IV secretion system protein VirD4 (TrsK) gene were highly expressed during the early exponential phase (5 h culture) of the host cells. However, most hypothetical genes scattered on the plasmid make it difficult to specify their role in the conjugation process.

### Loss of pBV71 during conversion from non-mucoid to mucoid phenotypes

During the cultivation of strain GH1-13 in TSB and R2A media at 25 °C, the non-mucoid phenotype of wild-type (WT) cells was often converted to the mucoid phenotype (Fig. 1a), as similar to that of a heat-cured strain[11]. Using 300 randomly isolated colonies, including 282 non-mucoids and 18 mucoids, PCR was performed with four primer sets targeting three genes (*prapD*, *traA*, and *traL*) in the plasmid and 16S rRNA gene in the chromosome to estimate the rate of loss of the plasmid during the non-mucoid to mucoid conversion under standard culture conditions. In preliminary tests, 282 non-mucoids were positive for all three plasmid genes and 18 mucoids were negative for all three plasmid genes. All colonies were positive for the 16S rDNA segment amplified by PCR. However, PCR using genomic DNA extracts showed that 16 of the 18 mucoids were positive for all three plasmid genes. Of the other two, one was positive for *prapD* and *traA* but negative for *traL*, and one was negative for all three plasmid genes (Fig. 1b). It was unclear whether false-negative (colony) or false-positive (genomic DNA) groups of the mucoid phenotype were related to the low plasmid DNA content, possibly either owing to a partial loss of plasmid or a mixed state of non-mucoid upon conversion to mucoid.

To examine the presence or absence of intact plasmid in each mucoid, PCR for the full-length plasmid genome was conducted using 20 primer pairs with head-to-tail and tail-to-head arrangements (Fig. 1c and Supplementary Table 1). The results showed that 16 mucoids positive for all target genes had the same PCR fragments as those of the plasmid genome of WT cells. Compared to these, the mucoid positive for *prapD* and *traA* but negative for *traL* showed a partial loss of the plasmid DNA between 46,551–50,330 bp (#14), 54,181–62,090 bp (#16 and #17), and 65,913–69,790 bp (#19), which explained the negative PCR result for the *traL* segment. PCR analysis also proved that the mucoid negative for all three plasmid target genes had lost almost all of the plasmid genome, which gave an estimated curing rate of 0.3% of the total population. Thus, although the number of mucoids is lower than the number of non-mucoids, it may be likely that a deficiency of pBV71 in a sub-population of host cells will present a mucoid phenotype.

### Variations in antibiotic resistance between non-mucoid and mucoid phenotypes with or without pBV71

The effects of mucoid phenotypes, with or without pBV71, on antibiotic resistance were compared with non-mucoid phenotypes of WT strains GH1-13 and FZB42. Among the tested antibiotics and toxic metal ions (Supplementary Table 2), mucoid GH1-13 variants with or without the plasmid became more sensitive to bacitracin (minimum inhibitory concentration [MIC] of 1.25 U/mL) and gramicidin (0.25 μg/mL), which target cell wall and cell membrane, respectively. They were also sensitive to selenite (2.5 μg/mL) and tellurite (8 μg/mL). Additionally, the MIC value of ampicillin was decreased in the mucoid variant (16 μg/mL) with an intact

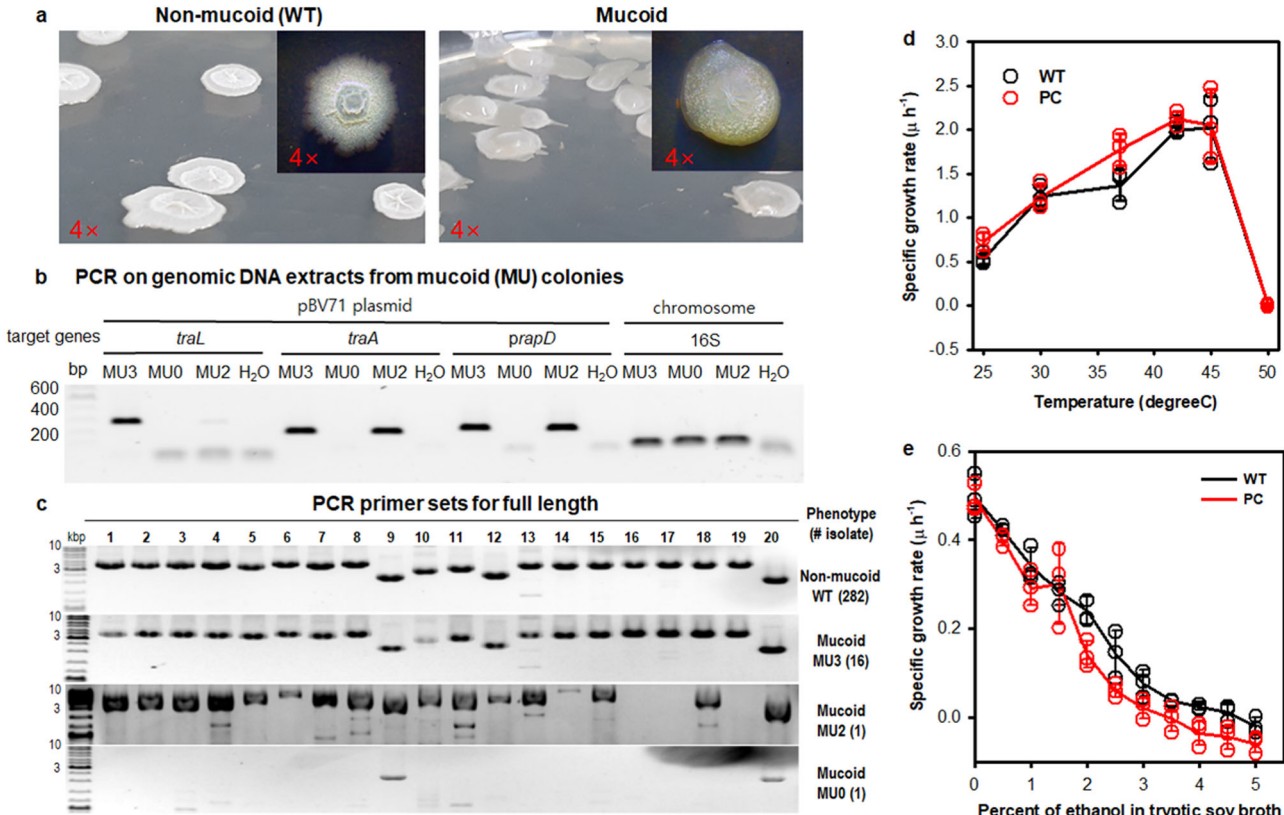

**Fig. 1 Conversion of non-mucoid to mucoid phenotype of *Bacillus velezensis* strain GH1-13. a** Colony morphology of non-mucoid (WT) and mucoid variants. WT strain GH1-13 shows as an irregular round, opaque white colony with umbonate elevation and wrinkled surface during cultivation on TSA and R2A plates at 25 °C, whereas mucoid variants form translucent white, convex, mucous colonies. **b** PCR screening of mucoid variants detected using genomic DNA extracts as templates and three plasmid target loci (*traL*, *traA*, and *prapD*) and a 16S ribosomal RNA gene segment as a genomic DNA control. **c** Evaluation of mucoid variants by PCR of the full-length plasmid genome is described in the "Methods" section. **d** Effects of temperature on the specific growth rate observed by measuring the OD600 during an early exponential phase in tryptic soy broth with aeration (180 rpm). **e** Effects of ethanol concentration (%, v/v) on the specific growth rate of cells in tryptic soy broth at 25 °C with aeration. $n = 3$ independent experiments (Supplementary Data 3). The mean and standard deviation (error bar) are plotted for all graphs.

plasmid when compared to that in the WT strain GH1-13 (128 µg/mL), while plasmid-cured (PC) cells reverted to ampicillin resistance. This indicates that ampicillin resistance varies between different variants of strain GH1-13 with or without pBV71. In contrast, a non-mucoid phenotype of strain FZB42 was moderately resistant to ampicillin (64 µg/mL) and similar bacitracin and selenite resistance as that of the non-mucoid phenotype of strain GH1-13 was observed. The strain FZB42 also showed a moderate resistance to gramicidin (32 µg/mL), and tellurite (8 µg/mL), which ranged between WT and mucoid GH1-13 variants.

Antibiotic sensitivities of strains GH1-13 and FZB42 with similar genetic backgrounds, but for the plasmid, appeared to vary depending on the phenotype conversion or plasmid presence. Using these characteristics, we developed strategies of complementation and conjugation with the native pBV71 since it had no selection marker. To investigate the potential selective effects of heat and ethanol that cause changes in cell membrane fluidity and membrane protein compositions[18,19], we analyzed growth rates of WT and PC cells of strain GH1-13 cultivated in various ethanol concentrations and at different temperatures (Fig. 1d, e and Supplementary Data 3). The two phenotypes showed similar growth rates under different conditions, indicating that neither the phenotype conversion nor the presence of pBV71 in strain GH1-13 directly influences cell membrane fluidity and permeability to heat and ethanol.

**Reverse complementation of pBV71 into PC cells**. To investigate the effect of pBV71 on the relationship between phenotype conversion and antibiotic resistance, supercoiled pBV71 was purified from WT cells of strain GH1-13 and used to transfect PC cells by electroporation. Complemented (COM) cells were then isolated by successive enrichment cultures of serial 10-fold dilutions with selenite-containing selective media (Fig. 2a). Stably transformed cells with intact plasmid caused the non-mucoid phenotype to form wrinkled colonies (Fig. 2c, d). The WT and COM cells exhibited similar resistance to selenite, tellurite, and bacitracin, and gramicidin (Fig. 2e, f and Supplementary Table 2). Under microscopy, both WT and COM cells, but not PC cells, were observed to have capsular pili in the extracellular matrix of pellicles (Fig. 3a, b). Transmission electron microscopy revealed high density of reduced tellurium metal deposited in both extracellular and intracellular compartments encapsulated in a thicker extracellular matrix in WT and COM cells when compared to PC cells (Fig. 3c). Therefore, it appeared that pBV71, in association with the formation of a non-mucoid phenotype, confers a selective advantage to the host cells, such as resistance to bacitracin and gramicidin and reduction of selenite and tellurite.

Variations in antibiotic resistance between non-mucoid and mucoid phenotypes may be related to the expression of the phage-shock proteins LiaIH and the synthesis of lipopeptide surfactin in the cell envelope and extracellular layers, which have a direct role

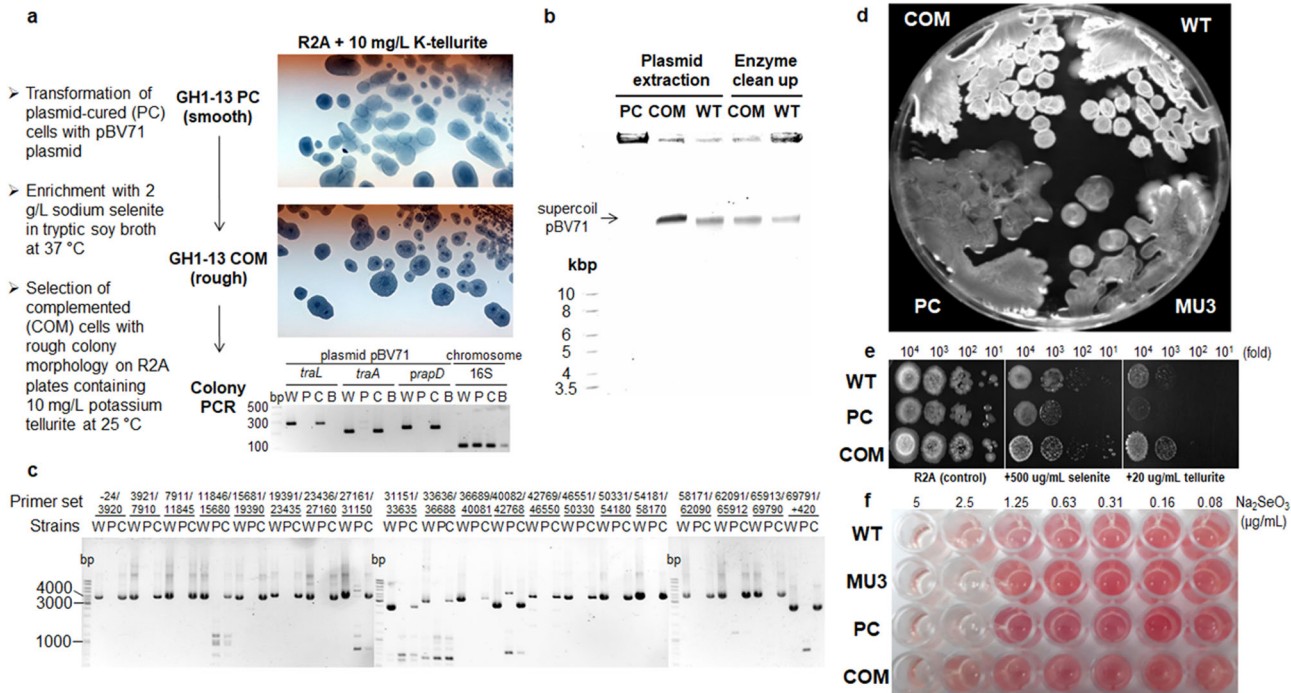

**Fig. 2 Complementation of pBV71 plasmid. a** Schematic description of the transformation of pBV71 into plasmid-cured (PC) cells and selection of a reverse complement COM after the enrichment culture with 2 g/L sodium selenite in tryptic soy broth with shaking at 37 °C. Smooth (mucous) and rough (umbonate) colony morphologies of PC and COM cells were observed after streaking the enriched culture repeatedly on R2A plates containing 10 mg/L potassium tellurite at 25 °C, and complementation of the plasmid was verified by colony PCR for three loci (*rapD*, *traA*, and *traL*) of the plasmid and part of the 16S rRNA gene as a control for genomic DNA. **b** Extraction and clean-up of supercoiled pBV71 plasmid used in the transformation into PC cells and the selection of COM cells, as described in the Methods section plasmid extraction and complementation. **c** PCR detection of the full-length plasmid DNA in genomic DNA extracts from wild-type (W), plasmid-cured (P), and complemented (C) cells by using 20 primer pairs for the head-to-tail and tail-to-head linkages as shown in Supplementary Table 1. **d** Colony morphologies of WT, mucoid variant (MU3) with pBV71, plasmid-free PC, and a reverse complement COM, showing the process of conversion to mucoid and reversion to non-mucoid phenotypes. **e** Spot dilution assays for WT, PC, and COM cells against 500 μg/mL sodium selenite and 20 μg/mL potassium tellurite in R2A agar plates. **f** Minimum inhibitory concentration of sodium selenite active to WT, MU3, PC, and COM cells. The cell growth is accompanied by the reduction of selenite to red selenium compounds.

in the resistance to bacitracin and gramicidin[20,21]. Additionally, the alterations in envelope sulfhydryl sites and thioredoxin disulfide reductase TrxR that control the adsorption and metabolism of selenite and tellurite, resulting in metal forms in the cells[22,23], may also contribute to the variation. Therefore, further analysis was required to examine the impact of pBV71 on the host cell response.

**Conjugation of pBV71 into strain FZB42 with a non-mucoid phenotype**. Strain FZB42 shows a non-mucoid phenotype with no plasmid, but it cannot create a mucoid phenotype observed in strain GH1-13. To measure the effect of pBV71 on antibiotic resistance of the non-mucoid phenotype, we used the strategy of conjugation to transfer pBV71 into strain FZB42 from a donor mucoid GH1-13 variant (MU3). MU3 is more sensitive to gramicidin, and hence, can be removed by overnight washout with a high dose of gramicidin above the MIC (200 μg/mL) in the TSB medium. In another experiment, when a high dose of ampicillin (128 μg/mL) was used in the co-culture, it failed to remove the donor and select a transconjugant. After treatment of the co-culture with gramicidin, a lethal dose (10 μg/mL) of tellurite in R2A plates increased the stable transconjugant population of FZB42 cells with intact pBV71, similar to that by reverse complementation.

Conjugation was validated by PCR for not only the full-length plasmid DNA, but also for three plasmid loci (*prapD*, *traA*, and *traL*) and three chromosomal loci of *yxaL*, Rap6 (BVH55_09210) and Rap12 (BVH55_14580), of which the tentatively assigned

Rap6 and Rap12 loci were only present in the genome of strain GH1-13, but not in the genome of strain FZB42. Therefore, the PCR products confirmed the removal of the donor and isolation of a transconjugant of the non-mucoid phenotype of strain FZB42 with the full plasmid DNA (Fig. 4a–c). When cells were incubated in TSB medium on a silicon wafer for scanning electron microscopy analysis, the donor and recipient showed similar images of cells, which were encapsulated in the extracellular matrix; however, the co-culture produced much longer pilus-like structures that connected cells across the loose extracellular matrix, possibly as a conjugation bridge for the transfer of plasmid DNA (Fig. 4d). The conjugation of FZB42 cells encased in the extracellular matrix was accompanied by a decrease in cell length ($1.7 ± 0.34$ μm) compared to that of the recipient (WT) cells ($2.97 ± 0.32$ μm).

The resulting transconjugant, FZB42p, possesses a similar non-mucoid phenotype as the WT strains GH1-13 and FZB42, which form wrinkled colonies on hydrophobic surfaces (Fig. 4e). However, its resistance to gramicidin (MIC > 128 μg/mL) and tellurite (MIC = 31 μg/mL), but not that to ampicillin (MIC = 16 μg/mL), is superior to that of the WT strain FZB42 and similar to that of another WT strain GH1-13 (Supplementary Table 2). These results reveal that the stable non-mucoid phenotype acquired by *B. velezensis* via conjugation of the pBV71 has an additional selective advantage over the non-mucoid phenotype of WT strain FZB42, whose antibiotic susceptibilities to gramicidin and tellurite range between that of the non-mucoid and mucoid phenotypes of strain GH1-13.

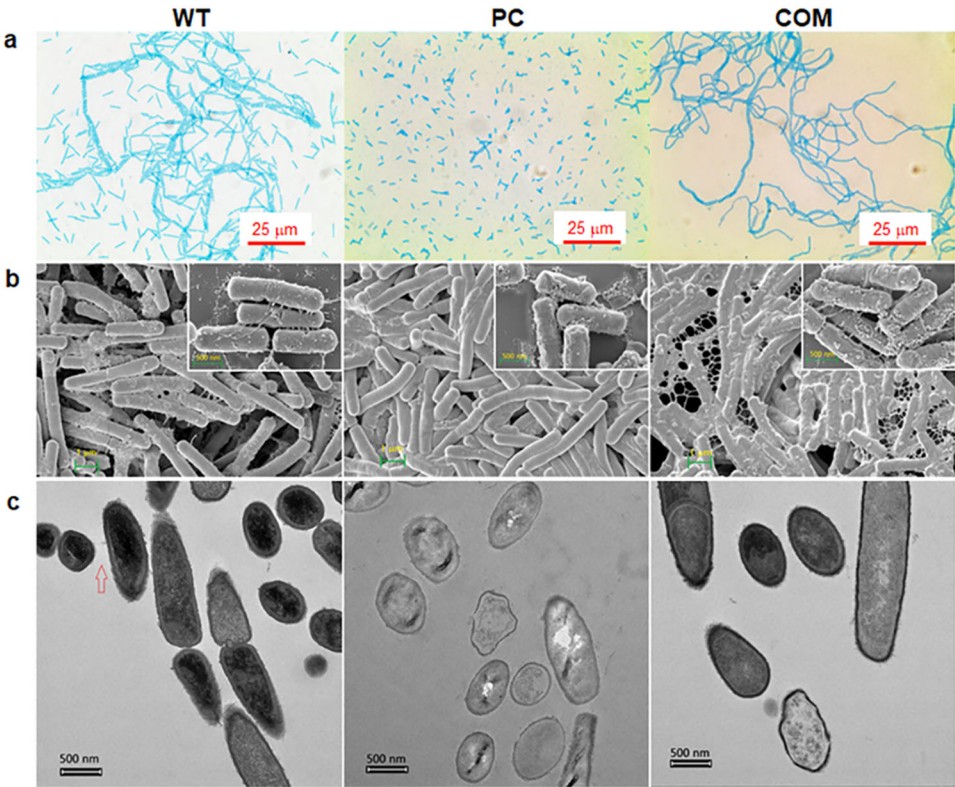

**Fig. 3 Morphologies of wild-type (WT), plasmid-cured (PC), and complemented (COM) cells of strain GH1-13. a** Microscopic images (1000-fold magnification) of WT, PC, and COM cells stained with Alcian blue for polysaccharide and Safranin O for cellular DNA. **b** Scanning electron microscopy (AURIGA) images of platinum-coated WT, PC, and COM cells at 20,000-fold magnification and 70,000-fold magnification (insets). **c** Bio-transmission electron microscopy images (Talos L120C, 120 kV) of the dissected WT, PC, and COM cells. The cells were treated with 10 mg/L potassium tellurite for 1 h during cultivation in tryptic soy broth at 25 °C with shaking (180 rpm). The head of the arrow shows a pilus-like structure that connects cells across the space.

**Alterations in growth, fermentation, and gene expression profiles of non-mucoid and mucoid phenotypes with or without pBV71.** Both non-mucoid and mucoid phenotypes of WT and mutant strains GH1-13 and FZB42 exhibited similar growth and fermentation profiles, except that the profiles of the mucoid phenotypes on lactose grew faster than that did the non-mucoid phenotypes, with or without pBV71 (Fig. 5a, b and Supplementary Fig. 3) and the PC cells that fermented D-tagatose under anaerobic conditions of API 50 CHB/E medium (Supplementary Table 3). Increased rates of lactose utilization by altering the substrate preference of mucoid phenotypes appeared independent of tagatose fermentation by the PC cells, as similarly observed in *B. licheniformis*[24]. These results indicated that the presence of pBV71 in strain GH1-13, which preferentially shows a non-mucoid phenotype, may cause catabolite repression of the lactose and tagatose utilization genes[25,26], although these genes have not yet been functionally defined in *B. velezensis*.

To explore the potential role of pBV71 in the linkage between phenotype change and catabolite control in the host, expression levels of 14 aspartate phosphatase Rap homologs, including the plasmid-encoded p*rapD*, were comparatively analyzed at early (5 h culture) and late (8 h culture) exponential phases of WT, PC, and COM cells using semi-quantitative RT-PCR (Supplementary Table 4). Among the analyzed Rap homologs, 13 genes were expressed in strain GH1-13, but the expression of a *rapC* ortholog (BVH55_02490) in the chromosome of strain GH1-13 was not detected (Supplementary Fig. 4). Expression levels of p*rapD* were high in the early exponential phase of WT and COM cells, but

not in PC cells compared to those in the late-exponential phase. The p*rapD* expression, predicted to be regulated by cognate sigma factors SigA and SigH, was altered with six chromosomal Rap homologs located in BVH55_02505 (*thrD*), BVH55_09210 (*rapF*2), BVH55_14245 (*lysC*), BVH55_14580 (*rapC1*), BVH55_18385 (*rapD*), and BVH55_18535 (*rapB*).

Moreover, RT-PCR results for QS systems (ComQXAP and Rap-Phr pairs) showed positive relationships between the expression of p*rapD* and chromosomal *rapF2–phrF2* genes in the early exponential phase of strain GH1-13 (Fig. 5c). In contrast, the early induction of p*rapD* expression in transconjugant FZB42p cells was accompanied by an upregulated *rapF* expression. During the exponential phase of cell growth, expression of *comA* that is transcriptionally regulated by a distinct Rap-Phr system[27] was not detected in any variant of strains GH1-13 an FZB42. Overall, these expression patterns appear to play a role in the QS-regulated expression of *rap* and *phr* genes involved in multiple and overlapping directions for cell specialization[28]. In the genome of strain GH-13, the PhrF2 sequence was not previously found owing to the usage of the alternative translation initiation site in the *rapF2* cassette and a lack of similarity between the propeptide sequence and any known sequences in gram-positive bacteria[29]. Manual curation of *rap–phr* genes showed that PhrF2 shares conserved N-terminal initiation sites and C-terminal peptide signal regions with a family of the other Rap–Phr pairs in strains GH1-13 and FZB42 (Fig. 5d). Although details of the mechanism remain to be determined, this analysis supports the hypothesis that a unique

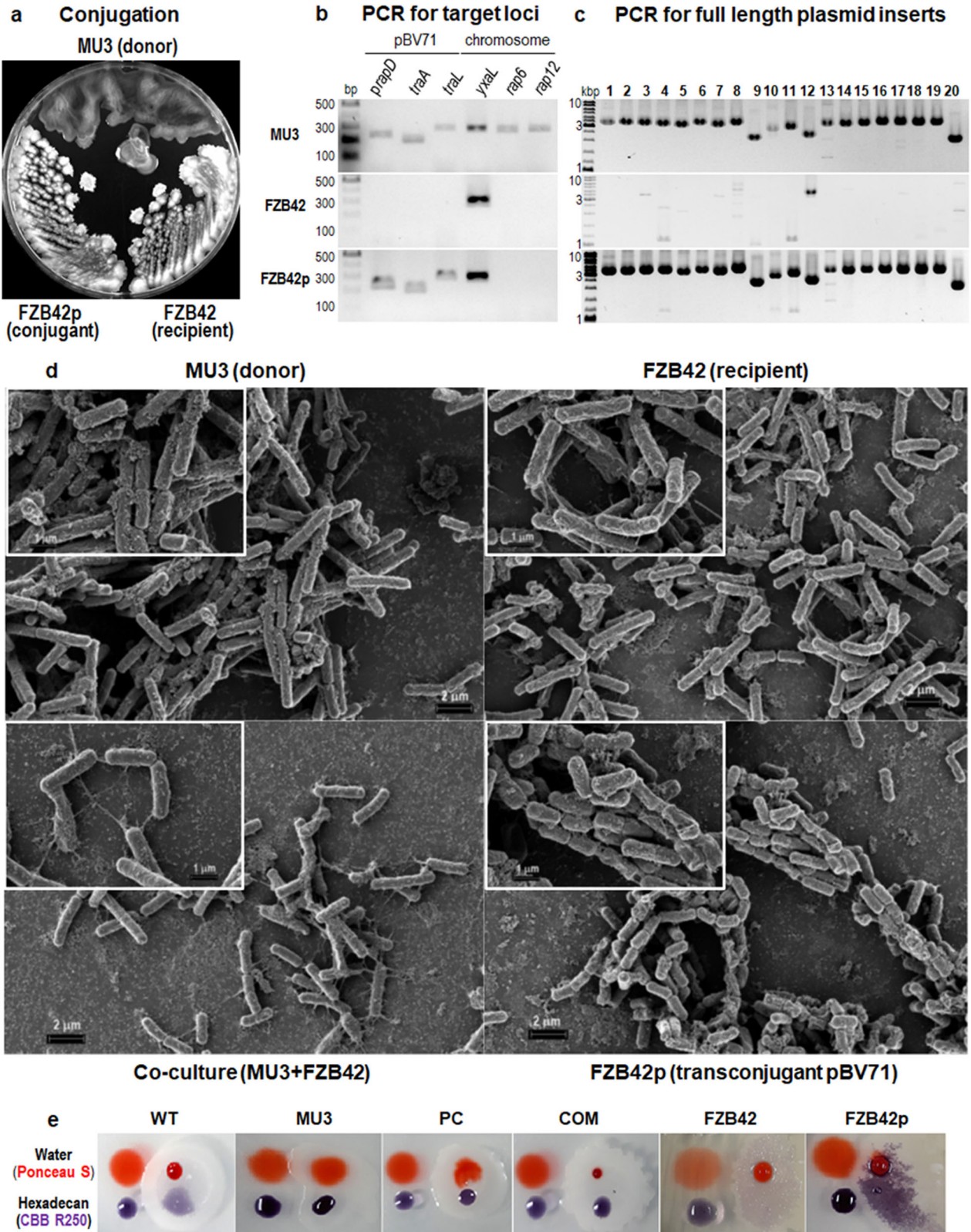

**Fig. 4 Conjugation of pBV71 into strain FZB42 from a mucoid variant (MU3) of strain GH1-13. a** Morphologies of MU3 (donor), FZB42 (recipient), and a transconjugant FZB42p with pBV71 on R2A plate. **b, c** PCR results for WT, MU3, and COM cells using plasmid and chromosome-specific primer pairs (**b**) and plasmid genome primer sets (**c**) as described in the Methods section. **d** Scanning electron microscopy (AURIGA) images of platinum-coated MU3 (donor), FZB42 (recipient), co-culture of donor and recipient, and a transconjugant FZB42p. Cells were pre-cultivated on a silicon wafer with TSB medium at 37 °C and fixed cell images were captured at 10,000-fold magnification and 30,000-fold magnification (insets). **e** Hydrophobicity tests for colony surfaces of WT, MU3, PC, COM, FZB42, and FZB42p cells grown on R2A plates. Hydrophobicity was detected on colony surfaces by the degree of absorption of 5 μL drops of water and hexadecane containing Ponceau S and Coomassie Brilliant Blue (CBB) R250 indicator dyes.

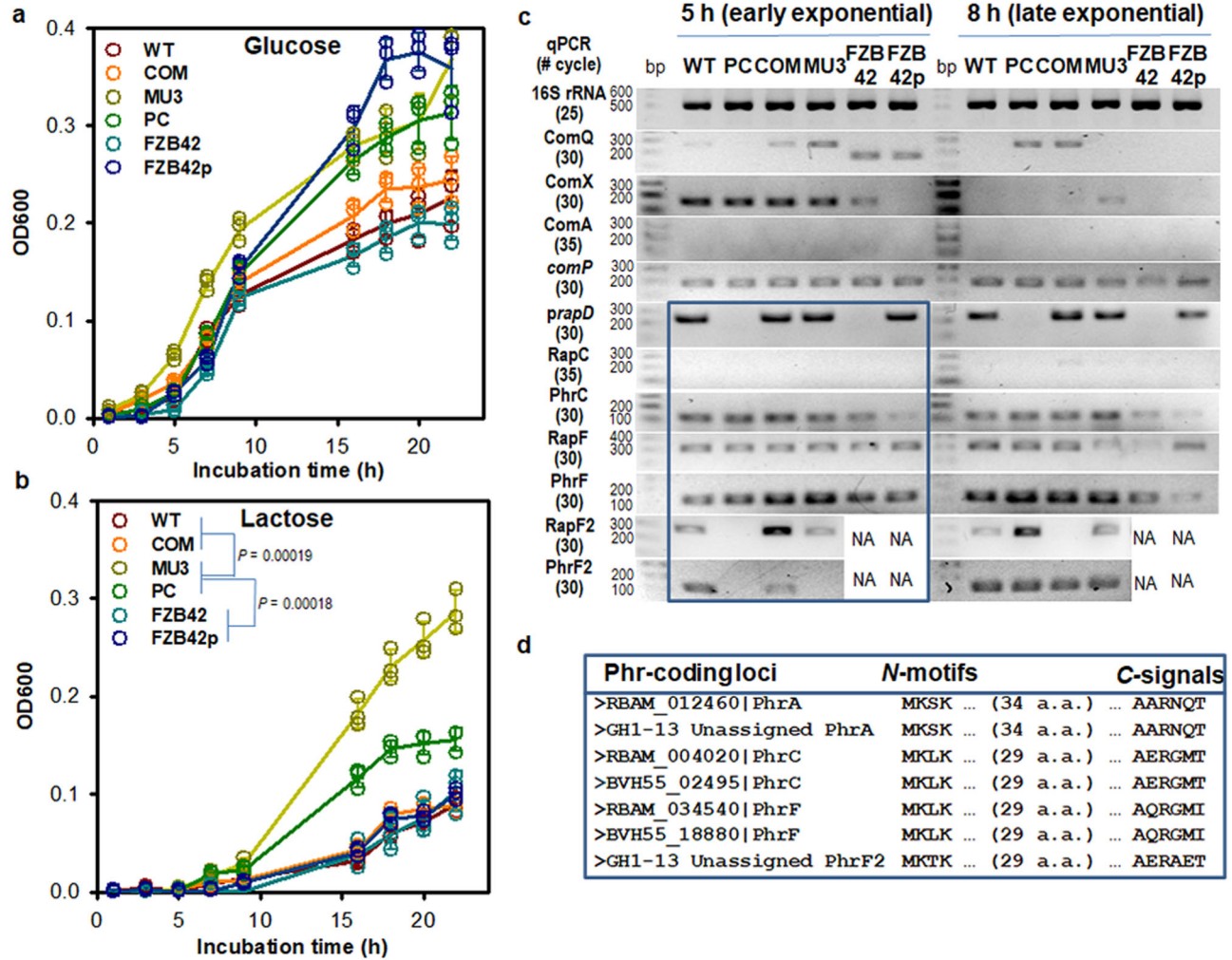

**Fig. 5 Growth curve and gene expression patterns of non-mucoid and mucoid phenotypes of strains GH1-13 and FZB42 with or without pBV71. a, b** Growth curves of WT strain GH1-13, mucoid variants MU3 and PC, a reverse complement COM, WT strain FZB42, and the transconjugant FZB42p cells on glucose (**a**) and lactose (**b**) at the final concentration of 0.2% (w/v) in M9 media. A statistically significant difference between the growth rates of cells of the non-mucoid and mucoid phenotypes with lactose used as the sole carbon and energy were determined by two-tailed $t$-tests with $P$-values < 0.05, from comparative tests for 6 sole carbon sources (glucose, lactose, maltose, sucrose, galactose, and glycerol) as shown in Supplementary Fig. 3. $n = 3$ independent experiments (Supplementary Data 3). The mean and standard deviation (error bar) are plotted for all graphs. **c** Semi-quantitative RT-PCR results show variations of the transcription of the quorum-sensing ComQXAP and Rap-Phr systems with expression levels of *prapD* in pBV71 in non-mucoid and mucoid variants of strains GH1-13 and FZB42 with or without pBV71. The uncropped gel images are shown in Supplementary Fig. 12. Expression levels of *prapD* and chromosomal *rap–phr* genes at the early exponential phase (5 h culture) of the cells are highlighted in the blue box. NA, *rapF2–phrF2* genes not available in strain FZB42. **d** Aligned amino acid sequences of *N*-terminal translation initiation sites and *C*-terminal pheromone peptide signals deduced from the coding regions of strains GH1-13 and FZB42. In strain GH1-13, PhrA, and PhrF2 sequences are categorized by a manual curation process.

RapF2–PhrF2 system may evolve incrementally into a QS system to manage the impact of pBV71 in strain GH1-13, but not in strain FZB42 and others.

**Assessment of early response to transcription of plasmid DNA in strain GH1-13.** Most Rap proteins have conserved roles in the inhibition of biofilm formation by preventing the phosphorylation of essential regulators, ComA, Spo0F, or DegU. These inhibitory activities are blocked by the binding of cognate Phr peptides. Plasmid-encoded Rap proteins in *Bacillus* species are believed to diversify the Rap–Phr systems that strictly control biofilm formation and sporulation and improve the adaptation capacities of the bacteria to environmental changes[30–36]. We observed that WT and COM cells containing the pBV71 began to form spores earlier than that did the PC cells at 40 h during aerobic starvation conditions in TSB medium, with no apparent

changes in growth rates of the cells (Fig. 6a). The sheathed WT cells were more efficient in increasing cell densities in air–liquid and water-sediment interfaces in static culture conditions than that were the PC cells (Fig. 6b). To assess the impact of pBV71 on early response regulators in strain GH1-13, RT-PCR was performed using primer sets targeting p*rapD* and plasmid conjugative transfer genes (*traA* and *traL*), as well as Rap target phosphorelay-associated genes during early and late-exponential phases in WT, PC, and COM cells (Supplementary Fig. 5). The results showed that the expression of the plasmid genes was accompanied by an upregulated expression of several Rap target genes, including *spo0F* and *degU*, in the early exponential phase of WT and COM cells but not in PC cells (Fig. 6c). These regulators appeared to associate with the early response to increased expression of pRapD.

In this case, increased pRapD that binds to a target protein or a cognate Phr peptide can jointly control a complex colony

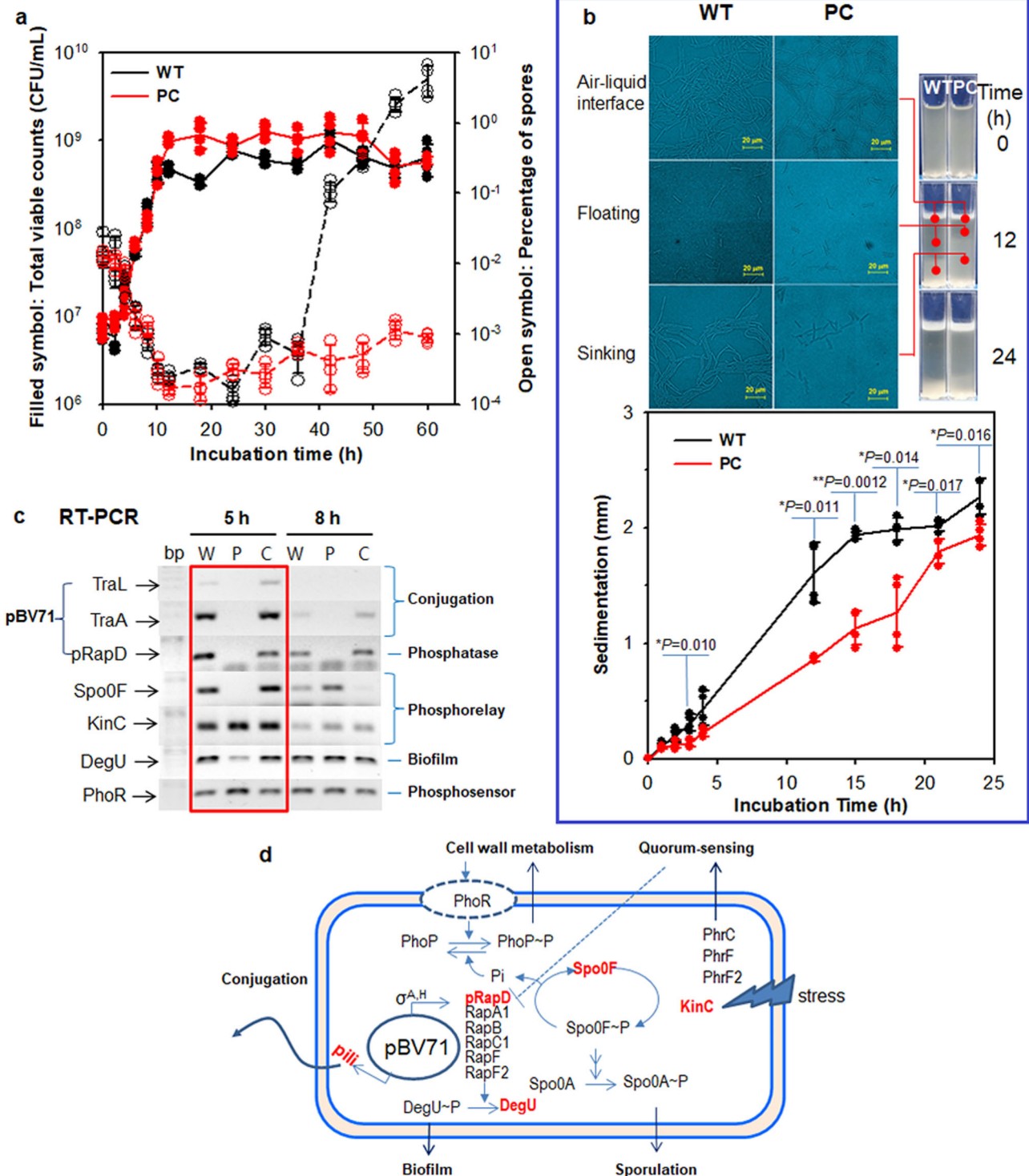

**Fig. 6 Growth characteristics and plasmid-associated gene expression patterns in non-mucoid and mucoid phenotypes of strain GH1-13 with or without pBV71. a** Growth curves (solid lines) and percentage of spores (red dotted lines) of WT and PC cells during cultivation in tryptic soy broth at 25 °C with shaking (180 rpm). **b** Variation in cell density and morphology in air–liquid interface, floating, and sinking cells during static culture of WT and PC cells in acryl cuvettes (10 × 10 × 40 mm) installed with transparency polypropylene slips. The variation in crystal violet-stained cell densities with depth along the slips is shown in the left photos. Statistically significant differences between sedimentation rates of cells were tested by two-tailed *t*-tests with *P*-values of <0.05. *n* = 4 independent experiments (Supplementary Data 3). The mean and standard deviation (error bar) are plotted for all graphs. **c** Semi-quantitative PCR results for three plasmid gene products (TraL, TraA, and pRapD) and associated response regulatory gene products (Spo0F, KinC, DegU, and PhoR) in WT, PC, and COM cells at 5 h (early exponential phase) and 8 h (late-exponential phase). The early response to transcription of the plasmid genes is highlighted in the red box. **d** Schematic illustration showing the impact of pBV71 on conjugation and biofilm formation of host cells in a quorum-sensing manner via a crosstalk between pRapD and chromosomal Rap-Phr systems.

development with co-expressed Rap-Phr systems[14], which reversely correlate with the expression of phosphate metabolism regulator PhoR[37], as schematically shown in Fig. 6d. The transcripts of pRapD and other types of Rap proteins sharply decrease with the down-regulation of Spo0F in the late-exponential phase of WT and COM cells, compared to that in the PC cells, whereas the PhrF2 expression increases in all non-mucoid and mucoid phenotypes (Fig. 5c). Moreover, this represents crosstalk between the plasmid and chromosome to form a bistable loop that controls conjugation in a quorum-sensing manner with respect to the plasmid conjugative transfer mucoid phenotype (MU3), which does not induce high levels of PhrF2 to pRapD in the early exponential phase.

**Effects of pBV71 on antibiotic synthesis.** Non-mucoid and mucoid phenotypes of strain GH1-13 showed different effects on the antagonism of *Fusarium solani* among tested plant pathogens (Fig. 7a-b and Supplementary Fig. 6). Analysis of antibiotics in the filtered culture fluids showed that WT and COM cells were able to produce higher amounts of macrolactins than did the PC cells, whereas late-exponential-phase PC cells produced higher amounts of surfactin compounds antagonistic to *F. solani*[38] than that did the WT and COM cells (Fig. 7c). The chromatographic peaks of macrolactins and surfactins detected in each sample by UV spectroscopy and mass spectrometry were assigned to mass data using chemical databases and normalized to the total peak area of the base peak chromatogram (Supplementary Fig. 7). Taken together, differential regulation of antibiotic synthesis in *Bacillus* may be attributable to the increased expression of Rap proteins along with the conjugative plasmid expression, which can prevent the phosphorylation of a transcriptional activator, such as ComA, of the surfactin (*srf*) genes[39,40]. However, the method used here cannot detect any other antibiotics synthesized by the rest of the orthologous gene clusters that encode lipopeptide synthetases and polyketide synthases in the strain GH1-13.

**Effects of pBV71 on protein secretion.** To investigate a potential change in protein secretion from the host cells by the conjugative plasmid expression, we compared secreted proteins from WT, PC, and COM cells at different growth phases: 5 h (early exponential), 8 h (late exponential), 12 h (early stationary), and 24 h (late stationary). WT and COM cells showed a similar increase in the levels of secreted proteins during late-exponential phase, compared to that by the PC cells (Fig. 8a). Tandem mass spectrometry analysis of secretome samples from 5 and 8 h cultured WT and PC cells was conducted in order to identify secreted proteins, focusing on the effect of pBV71 on the early response in the host. In this study, a total of 310 proteins were identified after filtering for more than two unique peptides and a false-discovery rate of <0.01 (Supplementary Data 4). Among them, 165 pairwise identified proteins were used to calculate fold changes using the normalized spectral abundance frequency (NSAF) scores, to be able to determine the differentially secreted proteins along with differences in the conjugative plasmid expression between early and late-exponential phases of the two phenotypes (Fig. 8b). The result shows that WT cells released higher levels (>3 on the $\log_2$-scale) of signal recognition particle-docking protein FtsY and an autolytic endopeptidase LytE in the early exponential phase, compared to those by the WT cells in the late-exponential phase and PC cells in early and late-exponential phases (Fig. 8c). The WT cells appeared to enhance the secretion of cell envelope proteins and phage proteins related to the interconnection between conjugation and biofilm formation with horizontal gene transfer[41]. Compared to that in WT cells, PC cells increased the secretion of cellular and membrane proteins by more than 8-fold higher in the early

exponential phase than that in the late-exponential phase, of which several proteins including phosphate acetyltransferase Pta, cold shock protein CspC, and cell division protein FtsZ are categorized as plant interaction proteins[10]. Although many differentially secreted proteins do not require a specific explanation of the protein functions and interactions, these changes explain that the conjugative plasmid expression in strain GH1-13 affects cell wall integrity and protein secretion during the early exponential phase of cell growth.

It was obscure that the absence of the plasmid in strain GH1-13 led to increased secretion (release) of various membrane and cytoplasmic proteins owing to a change in the membrane permeability caused by the conversion to mucoid phenotype. To evaluate the effect of pBV71 on the secretion pattern of known plant-interacting proteins, we performed western blotting for two extracellular beta-propeller proteins, 3-phytase Phy[4], and lateral root growth-promoting protein YxaL[12] (Fig. 8d). PC cells increased the secretion of Phy compared to that did the WT and COM cells in all four phases (two-tailed Z-test, $P < 0.05$), but there were no significant differences in the secretion pattern of YxaL among the three strains. These results are consistent with the constitutive expression of *yxaL*[12] and the phosphate starvation-inducible expression of *phyC* under the control of PhoPR[42]. Because the *phyC* expression is up- or downregulated by altering the level of phosphorylated PhoP~P through the regulation of PhoP-PhoR two-component system in *B. velezensis* FZB42[42], increased secretion of PhyC is strongly related to the upregulation of *phoR* expression in PC cells compared to that in the WT and COM cells in early exponential phase (Fig. 6c). This suggests that the conjugative plasmid expression affects the expression pattern of some PGPR traits as well as the physiological and morphological traits of the host.

**Integrative multi-omics analysis of wild-type and plasmid-cured cells of strain GH1-13.** Here, we analyzed transcriptome and proteome to access the role of pBV17 in the host cell. When examining transcriptome, all statistical analyses were performed using the TMM values (Supplementary Fig. 8). We estimated the variance of each gene in different phases of WT and PC cells and found that WT cells markedly increased the envelope stress-responsive *liaIH* expression with 99% confidence by more than 6.5$\log_2$-fold during the transition from early (5 h) to late (8 h) exponential phases, with a significant decrease in the expression of the pyrimidine metabolism *pyr* operon (Fig. 9a). A Yates-corrected chi-square test on different phases of WT and PC cells also showed that the presence of pBV71 in strain GH1-13 was associated with the *liaIH* expression ($\chi^2$-test, $P = 0.0011$–0.0002) rather than the *liaGFSR* expression ($\chi^2$-test, $P = 0.053$–0.523). These results concurred partly with the different expression patterns of *liaSR* and *liaIH* as shown by semi-quantitative RT-PCR (Supplementary Fig. 5), indicating a differential effect of pBV71 on early *liaGFSR* and late *liaIH* responses. These responses are possibly mediated by a heterogeneous gene expression or antibiotics produced by the bacterium itself[43–45]. Post hoc Pearson's correlation analysis on different phases of WT cells showed that the *liaIH* response positively correlated with additional 89 genes and three major sigma factors, *sigA*, *sigH*, and *sigW*, of which many genes were differentially expressed by more than 6.5$\log_2$-fold in late-exponential phase (Supplementary Data 5). This rigorous change was different from that of PC cells in the exponential phase, but similar to them in the stationary phase.

Time-coursed proteome analysis showed that LiaH levels were higher in WT cells than those in PC cells, and its levels were positively correlated ($r > 0.99$) over time with the levels of

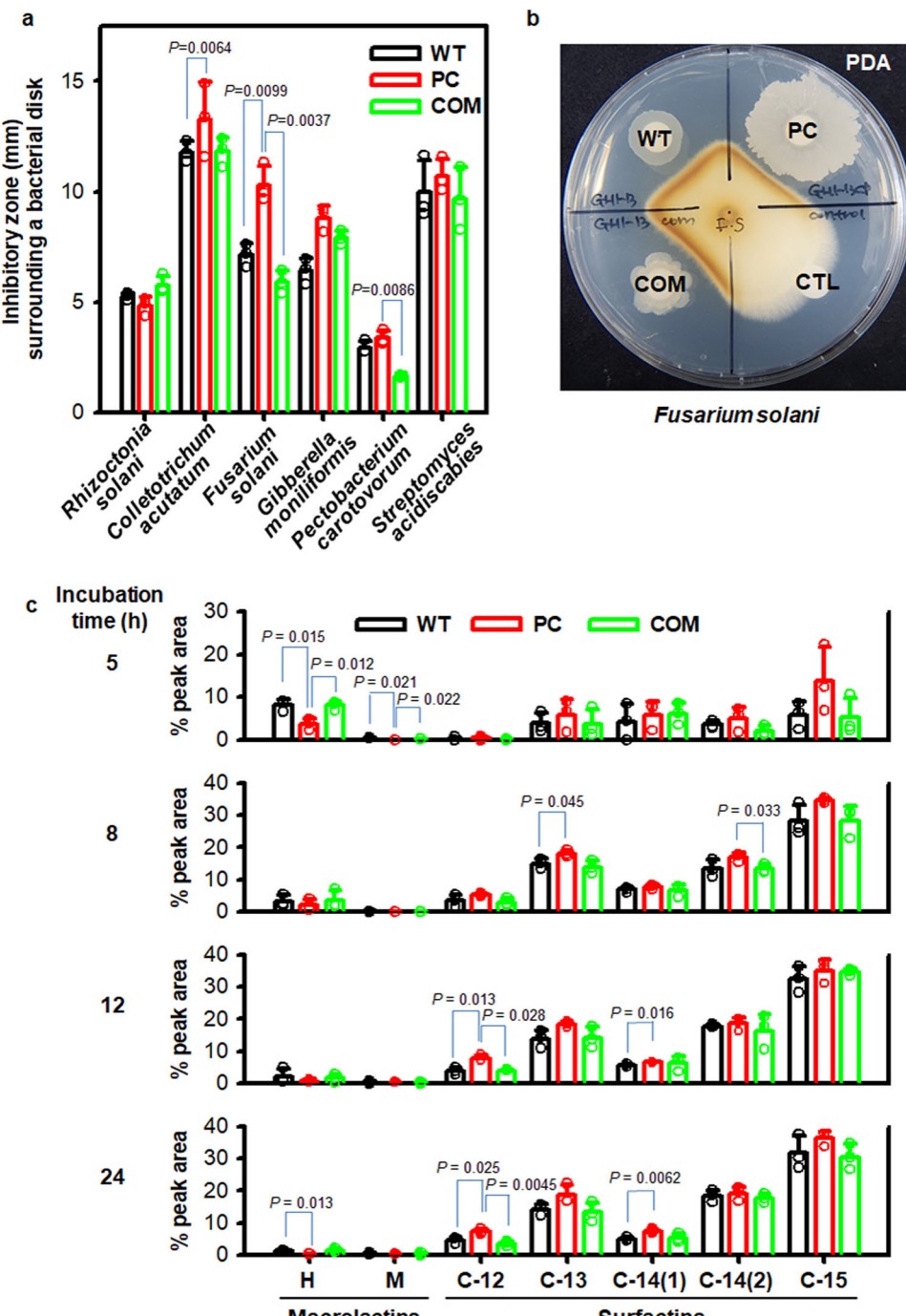

**Fig. 7 Antagonistic test and analysis of antibiotics released from strain GH1-13. a** In vitro inhibition of plant pathogens tested as shown in Supplementary Fig. 6. **b** A typical photo of *Fusarium solani* tested in vitro by bacterial disk assays of wild-type (WT or W), plasmid-cured (PC or P), and complemented (COM or C) cells on a potato dextrose agar (PDA) plate. **c** Levels of macrolactins (**H** and **M**) and surfactins (**C12** to **C15**) in the filtered culture fluids when cells were grown in tryptic soy broth at 25 °C and 180 rpm. The chemical properties and structures determined by HPLC with UV spectroscopy and mass spectrometry are shown in Supplementary Fig. 7. Statistically significant differences in the percentage of each peak area between strains at the incubation times of 5, 8, 12, and 24 h were determined by two-tailed *t*-tests with *P*-values of less than 0.05. *n* = 3 independent experiments (Supplementary Data 3). The mean and standard deviation (error bar) are plotted for all graphs.

polyketide synthases, BaeM, BaeR, and MlnI (Fig. 9b and Supplementary Data 6). The analyzed proteome moderately correlated with the corresponding secretome, but the dynamic changes of protein levels both in proteome and secretome little correlated with the corresponding gene expression changes (Supplementary Fig. 9). Although mRNAs and proteins have many different types of regulation of half-lives as they vary from time to time, a combined analysis was conducted using STRING[46] to obtain insight into the functional association networks of pBV71 in host cells (Supplementary Fig. 10). The plasmid-induced *liaIH* expression was highly associated ($\chi^2$-test, $P < 0.0001$) with the activation of maltose metabolism *glv* operon (*malA-glvR-malP*) and various sugar transport PTS systems (*galK1*, *lacE1*, *licBCA*, *manP*, *mtlAF*, *treP*, and *ywbA*) under

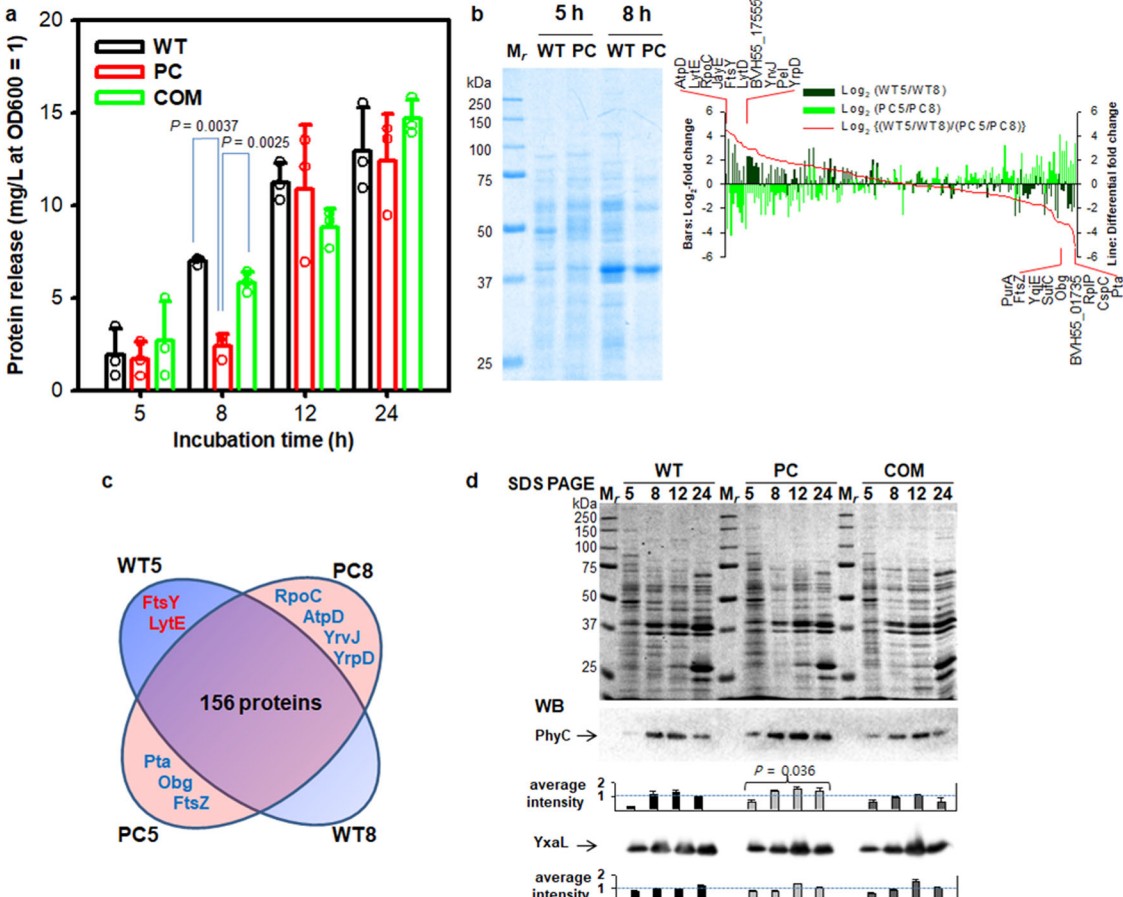

**Fig. 8 Comparison of proteins released from wild-type and plasmid-cured cells of strain GH1-13. a** Protein levels released at different growth phases. *P*-values at the time of 8 h were determined by two-tailed *t*-tests. *n* = 3 independent experiments (Supplementary Data 3). The mean and standard deviation (error bar) are plotted for all graphs. **b** SDS PAGE and tandem mass spectrometry data analysis of proteins in filtered (concentrated) 0.5 mL culture fluid per lane. Bar graphs show an inverse relationship of log₂-fold changes in the relative protein levels from the two different phases (5 h vs 8 h) of wild-type (WT) and plasmid-cured (PC) cells, when 165 protein entities were determined by the non-zero normalized spectral abundance frequency (NSAF) values from the pooled tandem mass spectrometry data in Supplementary Data 4. Significant differences in the log₂-fold changes between the two phases of WT and PC cells are shown by differential fold change cutoff to 3log₂-fold on both sides. **c** Venn diagram of differentially released proteins by >3log₂-fold among 165 proteins (Supplementary Data 4) detected in early (5 h) and late (8 h) exponential phases of WT and PC cells. **d** SDS PAGE of extracellular proteins and western blots for extracellular beta-propeller proteins Phy and YxaL in concentrated 0.5 mL culture fluids over time. Significantly high levels of Phy from PC cells were determined by a two-tailed *Z*-test on average intensity *μ* = 1. Full images of the western blots and Coomassie blue-stained membranes are shown in Supplementary Fig. 12.

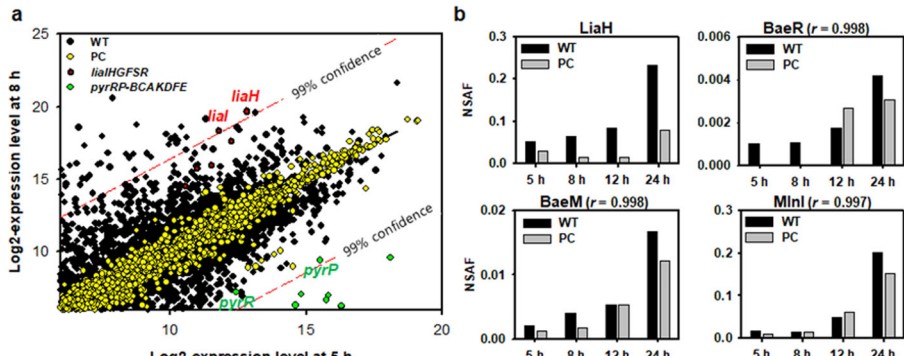

**Fig. 9 Differential gene expression in the exponential phase of *Bacillus velezensis* strain GH1-13. a** Plots of log₂ normalized RNA-Seq data for the early (5 h) and late (8 h) exponential phases of wild-type (WT) pBV71 plasmid-containing and plasmid-cured (PC) cells, showing the significant increase of *lialH* transcripts with 99% confidence by >6.5log₂-fold in the late-exponential phase of WT cells, with a significant decrease of *pyr* operon transcripts. **b** Normalized spectral abundance frequency (NSAF) ratios of proteins in whole-cell lysates from the different incubation times of WT and PC cells in Supplementary Data 6, showing significantly high levels of the pBV71 plasmid-induced LiaH protein compared to that of PC cells (two-tailed *Z*-test, *P* < 10⁻⁶) with strong correlation to the levels of polyketide synthases BaeM, BaeR, and MlnI determined by post hoc tests (Pearson's coefficient, *r* > 0.99).

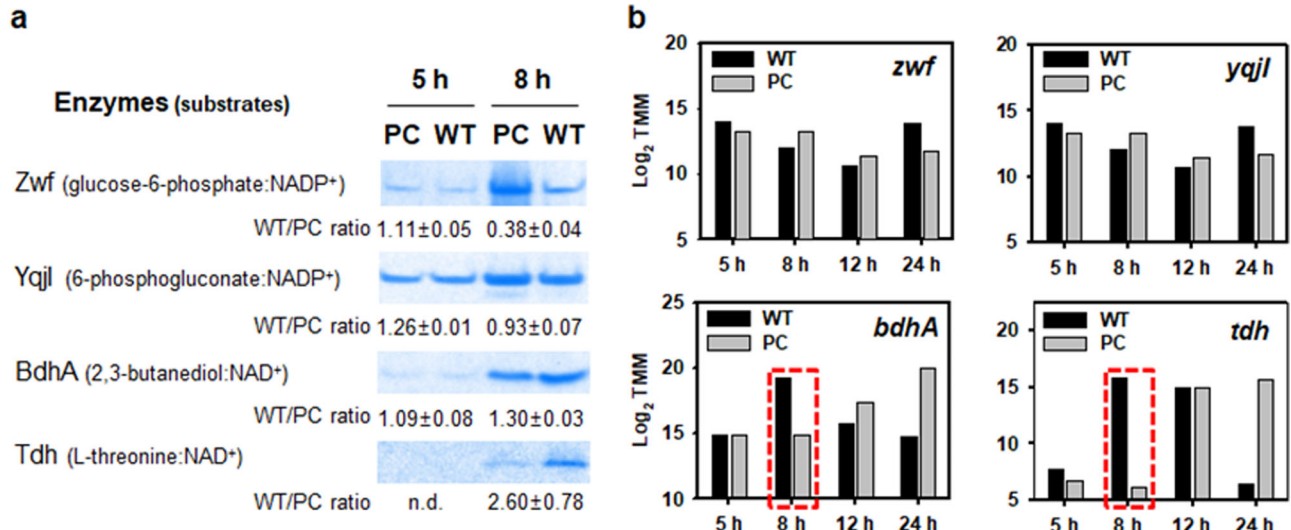

**Fig. 10 In-gel activities and gene expression levels of NAD(P)-dependent dehydrogenases. a** In-gel activity assays for NAD(P)$^+$-dependent enzymes toward glucose-6-phosphate (Zwf), 6-phophogluconate (YqjI), 2,3-butanediol (BdhA), and L-threonine (Tdh) in whole-cell extracts (20 μg protein per lane) obtained from early (5 h) and late (8 h) exponential phases of WT and PC cells by non-denaturing blue native polyacrylamide gel electrophoresis. Full images of the blue native gels are shown in Supplementary Fig. 12. **b** Log$_2$ transformed mRNA levels show the upregulation of *bdhA* and *tdh* expression in the late (8 h) exponential phase of WT cells as highlighted in a red dotted line box (Supplementary Fig. 11).

catabolite control[47,48]. The plasmid-induced PTS expression was opposed to the main PTS systems (*ptsG*, *ptsHI*, and *fruA*) for sucrose utilization[49,50], affecting levan synthesis in capsule or biofilm matrix[51,52]. This control facilitates extracellular polysaccharide biosynthesis with upregulation of the *eps* operon[53,54], amyloid biofilm protein TasA, and hydrophobic surface protein BslA (which allows it to form a more wrinkled colony with an increased surface and water-repellent properties[55–57]), as well as antimicrobial gene expression[58] including the upregulation of bacillaene (*bae*), bacillomycin (*bmy*), difficidin (*dfn*), fengycin (*fen*), and macrolactin (*mln*) genes, but not of surfactin (*srf*) genes. The proposed pathways reflect that the conjugative pBV71 affects metabolic and protective functions of the host cell in the exponential phase.

Finally, based on the results of combined transcriptome and proteome analysis, we measured *in-gel* activities of NAD(P)-dependent oxidoreductases to show that the metabolic pathways of 2,3-butanediol and L-threonine play a central role in the linkage between the carbon-energy metabolism (sugar transport PTS and metabolism, TCA cycle, and enoyl-CoA synthesis) and the antibiotic biosynthesis (Supplementary Fig. 11). Figure 10 highlights that NAD$^+$-dependent 2,3-butanediol dehydrogenase (BdhA) and L-threonine dehydrogenase (Tdh) activities markedly increased in WT cells in accordance with upregulated expression of the corresponding genes in the late-exponential phase. At this sampling time, PC cells showed slightly higher activities of NADP$^+$-dependent glucose-6-phosphate (Zwf) and 6-phosphogluconate (YqjI) dehydrogenases than those in WT cells. The overall results suggest that the plasmid-induced expression of BdhA and Tdh in the late-exponential phase of *B. velezensis* strain GH1-13 will have a pivotal role in maintaining homeostasis of cellular NADH and NAD$^+$ as a critical parameter in numerous bacteria that controls colony wrinkling[59–61], 2,3-butanediol synthesis[8,62], antibiotic synthesis[63–65], and resistance development[66].

In conclusion, pBV71 affects many PGPR traits of the host *B. velezensis* strain GH1-13. In this paper, we address the impact of pBV71 on the conversion from non-mucoid to mucoid phenotype and antibiotic resistance by complementation of the plasmid with no selective marker into PC cells sensitive to

bacitracin, gramicidin, selenite, and tellurite, and also by conjugation of the plasmid into strain FZB42 from a mucoid variant (MU3) sensitive to gramicidin and tellurite. Tellurite was effective in isolating a reverse complement of PC cells and a transconjugant of strain FZB42 with the native pBV71. Comparison of various phenotypes of different strains demonstrated that early transcription of pBV71 in the host cell affects conjugation, biofilm formation, and antibiotic production in a quorum-sensing manner via crosstalk between the plasmid-encoded orphan pRapD protein and chromosomal Rap–Phr systems, which occur widely in *Bacillus* species. Further, the multi-omics analysis revealed that the conjugative plasmid expression in the early exponential phase of host cells elicits the induction of *liaIH* genes involved in the cell envelope stress response and antibiotic resistance for cell survival and turns on the redox-regulatory mechanisms linked to the production of extracellular polysaccharide matrix, antibiotics, and wrinkled hydrophobic colony surfaces for protection of the host cells. Our study demonstrates that the non-mucoid phenotype of strain GH1-13 harboring the conjugative pBV71 has a selective advantage to improve cell differentiation and resistance to antibiotics and toxic metal ions better than that by the mucoid phenotypes deficient or lacking the plasmid, or another strain without cognate Phr.

## Methods

**Strains and cultivation.** A frozen stock of *B. velezensis* strain GH1-13 in 50% glycerol at −80 °C was thawed and cultivated in TSB medium or on TSA or R2A plates (BD, Sparks, USA). WT strain GH1-13 was repeatedly cultured for 5 cycles of exponential growth (5 h cultures) with aeration (180 rpm) in TSB medium at 25 °C, and serial 10-fold dilutions of the culture medium were plated on R2A plates. After 24 h incubation of the plates at 25 °C, 300 single colonies were randomly selected in order to determine their phenotypes, and were inoculated into 200 μL TSB medium and cultivated for 5 h at 25 °C to extract genomic DNA for PCR analysis. Mucoid variants with or without a full plasmid were isolated for complementation and conjugation experiments. For conjugation, *B. velezensis* strain FZB42 was obtained from the Korean Agricultural Culture Collection (KACC). All strains were grown in TSB with aeration (180 rpm) at 25 °C and synchronous growth was maintained by repeated 5 h culture at the initial optical density at 600 nm (OD600) of 0.01. Percentages of endospores at different incubation times were determined before and after heat treatment at 80 °C for 15 min by a serial 10-fold dilution method to determine colony-forming units (cfu) on TSA plates.

**Analysis of cell morphology**. When OD600 values of wild-type, plasmid-cured, and complemented strains reached ~1.0 (at 6 h incubation time) in normal TSB with aeration (180 rpm) at 25 °C, cell smears on glass slides were made, fixed with methanol, and heated at 60 °C heat for 15 min. The fixed cells were counterstained with Alcian blue and Safranin O which bind to extracellular polysaccharides and cellular DNA, respectively, and observed under an Olympus BX51 microscope installed with an IMTcam3-PLUS camera. At the end of the exponential cell growth at OD600 of ~1.0, 3 mL culture medium was transferred into sterile acryl cuvettes (10 × 10 × 40 mm) installed with sterile 3 M PP2900 polypropylene films (9 × 40 mm) at 5° slant from vertical to bottom in a static environment, and incubated at 25 °C for 12 h to observe the morphology of attached cells in the depth of air–liquid interface, floating, and sediment areas. The wet sample on coverslip was fixed with 2.5% (v/v) formaldehyde in 1×PBS (8 g NaCl, 0.2 g KCl, 1.44 g Na$_2$HPO$_4$, 0.24 g KH$_2$PO$_4$ in 1 L water) for 1 h, rinsed twice with 1 × PBS, and fixed with a drop of mounting medium (Dako North America, Carpinteria CA, USA) on glass slides. To conduct electronic microscopy in and on cells with reduced tellurite particles, exponentially growing cells in TSB at 25 °C were treated with 10 µg/mL potassium tellurite hydrate (purity >97%, Daejung Chemicals and Metals, South Korea) for 1 h and fixed with 2.5% glutaraldehyde for 24 h at 4 °C. The cells were post-fixed with 1% osmium tetroxide (Electron Microscopy Sciences, Hatfield PA, USA) in 50 mM sodium cacodylate and subjected to en bloc staining with 0.5% uranyl acetate and polymerization using Spurr's resin kit (Electron Microscopy Sciences, Hatfield PA, USA) followed by ultramicrotome sectioning of 70 nm thickness and bio-transmission electron microscopy using a Talos L120C instrument at 120 kV. For 3D analysis of encapsulated cells using a Carl Zeiss field-emission scanning electronic microscope (AURIGA), post-fixed cells were dehydrated gradually with ethanol, dried with hexamethyldisilazane, and coated with platinum in a Leica EM ACE200 low vacuum coater. All of the raw data of the microscopic images were deposited in the BioStudies database[67] under the accession number S-BSST608.

**In vitro antibiotic susceptibility tests and effects of temperature and ethanol**. The minimum inhibitory concentration (MIC) was determined by inoculation of about 10$^4$ CFUs of exponentially growing cells into 100 µL TSB in each well of 96-well microplates containing serial 2-fold dilutions of antibiotics and toxic metal ions given in Supplementary Table 2. Tightly sealed microplates were incubated with shaking (180 rpm) at 25 °C for 24 h after which the lowest concentration of each antibiotic agent that inhibited growth of bacteria was determined as the MIC. Spot dilution assays for resistance to selenite and tellurite were performed on R2A plates containing 500 µg/mL sodium selenite (purity ~98%, Sigma-Aldrich) or 20 µg/mL potassium tellurite hydrate. To examine the potential selective effect of heat and ethanol, specific growth rate of cells was measured by determining OD600 during cultivation in TSB with various concentrations of ethanol (0–5%, v/v) and at different temperatures in a range of 25–45 °C. Spot dilution assays were performed in a range of sodium selenite (0–10 mg/mL), and potassium tellurite hydrate (0–1 mg/mL).

**In vitro antagonistic tests against plant pathogens**. Antagonistic tests of wild-type, plasmid-cured, and complemented strains cultivated in TSB at 25 °C were performed against the following plant pathogens provided by the Korean Agricultural Culture Collection (KACC): *Rhizoctonia solani* KACC 40123 causing sheath blight of rice (*Oryza sativa*), *Colletotrichum acutatum* KACC 40804 causing anthracnose of pepper (*Capsicum annuum*), *Fusarium solani* KACC 44891 causing root rot of ginseng (*Panax ginseng*), *Gibberella moniliformis* KACC 44022 causing bakanae disease of rice, *Pectobacterium carotovorum* KACC 11130 causing soft root disease of ginseng, and *Streptomyces acidiscabies* S70 causing scab lesion on potato. Fungal cultures (*R. solani*, *C. acutatum*, *F. solani*, and *G. moniliformis*) and bacterial cultures (*P. carotovorum* and *S. acidiscabies*) were maintained and tested, respectively, on BD potato dextrose agar (PDA) and TSA plates at 25 °C. Each 5 µL TSB medium of exponentially growing cells of a test strain at OD600 of ~1.0 was inoculated onto sterile 3 M filter paper disks (diameter, 6 mm) and placed on the corners of PDA plates inoculated with the four fungal cultures in the center or in plates with lawns of the two bacterial cell cultures. Control disks soaked only with 5 µL TSB were placed on one corner of each plate. After incubation for 3–5 days at 25 °C, the zone of growth inhibition of the pathogen around the TSB culture medium disk was compared to the control disk measured from the disk center to inhibition edge with scale in mm.

**PCR analysis**. Using 300 randomly selected single colonies, the presence of pBV71 in the colony or genomic DNA extract was detected by PCR with primer pairs for *prapD* (locus-tag BVH55_00015: forward 5′-GCGTTAAGAACACCTACAGA-3′ and reverse 5′-AATGCATTCTCAGCGTGG-3′), *traA* (locus-tag BVH55_00285: forward 5′-ACCTTCACATGCTCCTGA-3′ and reverse 5′-TGTATTGAAACGTC CGCAAC-3′), and *traL* (locus-tag BVH55_00430: forward 5′-TGCGGCTGTAAA AGATGGA-3′ and reverse 5′-GGCAGGTTACATCAATCCA-3′) in the plasmid genome sequence (GenBank accession number CP019039.1). The genomic DNA was extracted using Wizard DNA purification kit (Promega, Madison WI, USA). As a chromosomal reference sequence, a portion of 16S rRNA gene was amplified using a universal primer pair 27F-519R (forward 5′-AGAGTTTGATCCTGGCT CAG-3′ and reverse 5′-GTATTACCGCGGCTGCTG-3′) or a *Bacillus* 16S rRNA

gene-specific primer set (forward 5′-CCTACGGGAGGCAGCAGTAG-3′ and reverse 5′-CAACAGAGCTTTACGATCCGAAA-3′)[12]. Quantitative PCR was performed using GoTaq qPCR Master Mix (Promega, Madison, USA) incorporating a hot start at 95 °C for 2 min, followed by 30–35 cycles of melting at 94 °C for 15 s, annealing at 58 °C for 15 s, and elongation at 72 °C for 30 s. For full-length plasmid fingerprinting, DNA inserts were amplified by increasing elongation at 72 °C for 3 min during 25 cycles of a PCR run using Takaka ExTaq mixture with addition of 5% dimethylsulfoxide, genomic DNA extracts as templates, and each set of 20 primer pairs for the head-to-tail and tail-to-head linkages. The PCR primers used are shown in Supplementary Tables 1 and 4.

**Plasmid extraction**. Supercoiled pBV71 was extracted from the original strain GH1-13 by a method modified from a previous procedure[68]. Briefly, cells were harvested after repeated 5 h culture corresponding to early exponential phase in 5 mL TSB with aeration (180 rpm) at 25 °C and mixed thoroughly with lysing buffer containing 3% sodium dodecylsulfate (SDS) and 50 mM Tris, whose pH was adjusted to 12.6 with 2 M NaOH. The buffer temperature was pre-warmed to 60 °C before used. After the lysis of cells at 60 °C for 15 min, the lysate was gently mixed with an equal volume of 1% low-melting-point (LMP) agarose (Gibco BRL, Gaithersburg MD, USA), pre-warmed at 60 °C, and then extracted with an equal volume of phenol-chloroform-isoamyl alcohol (25:24:1, v/v/v) inverting the tube several times. During centrifugation at 16,000 × g for 5 min, LMP agarose was solidified to separate a buffy layer of chromosomal DNA and high-density molecules, whereby a clear supernatant of plasmid DNA was obtained. The DNA concentration was determined by a DeNovix DS-11FX Spectrophotometer (DeNovix, Wilmington DE, USA). Then, the extract was cleaned up with a mixture of RNase H, BamH1, EagI, EcoR1, SmaI, PstI, and XhoI restriction enzymes in 1 × buffer Tango (Fermentas, Burlington ON, Canada) at 37 °C for 1 h in order to purify the plasmid DNA with no target sites. This step was not essential for complementation. Rapidly extracted and enzyme-cleaned plasmids were visualized by electrophoresis using a 0.5 % SeaKem GTG agarose (Lonza, Rockland MD, USA) gel in 1 × TAE buffer (40 mM Tris, 20 mM acetic acid, and 1 mM EDTA, pH 8) at 25 V at 15 °C, followed by staining with ethidium bromide.

**Complementation of the native pBV71**. Electrocompetent PC cells were prepared from repeated 5 h culture in 5 mL TSB with aeration (180 rpm) at 25 °C. After chilling in ice water for 30 min, cells were centrifuged at 3500 × g and 4 °C for 5 min, and washed twice with ice-cold 1 × TE buffer (10 mM Tris and 1 mM EDTA, pH 8), and twice with ice-cold D.W. Approximately 10$^8$ cells (1 mL of OD600 ~0.5) were suspended in 50 µL of 10% glycerol, mixed with 100 ng plasmid DNA, and incubated on ice for 30 min. Electroporation was performed using a BioRad MicroPulser at 1.8 kV for 2.5 ms. Electroporesed cells were suspended in a pre-warmed 1 mL SOC medium and incubated according to the laboratory manual[69]. Then, cells were aerobically cultivated at 37 °C for 5 h in 2 × YT medium (10 g yeast extract and 10 g tryptone in 1 L D.W.) supplemented with 40 mM glucose and 2 mg/mL sodium selenite. During successive cultures of the transformant in selenite-containing selective medium, 10 µL portions were streaked on R2A plates containing 10 µg/mL potassium tellurite hydrate. The reverse complement was clearly distinguished by the wrinkled colony morphology, showing black crystals of the reduced tellurium or amorphous regions of the red selenium compounds. Complementation was confirmed by PCR analysis.

**Conjugation of the native pBV71**. A mucoid variant (MU3) of strain GH1-13 with pBV71 and strain FZB42 with no plasmid were used as donor and recipient of pBV71. Cells were exponentially grown in TSB medium at 37 °C and 180 rpm for 3 h, harvested by centrifugation at 3500 × g and 4 °C for 5 min, and washed twice with ice-cold 1 × TE buffer (10 mM Tris and 1 mM EDTA, pH 8), and twice with ice-cold D.W. Approximately 10$^9$ cells (cfu) of each strain were mixed and incubated on a sterile 0.2 µm-membrane filter, Ø 47 mm (Satorius, Goettingen, Germany) on a TSA plate for 5 h at 37 °C. The co-culture on the membrane was transferred into 2 mL TSB medium containing 200 µg/mL gramicidin overnight with shaking (180 rpm) at 37 °C to remove the donor cells. The gramicidin-treated cells were 10-fold serially diluted and plated on R2A plates containing 10 µg/mL potassium tellurite to isolate a transconjugant of strain FZB42 with the pBV71 by PCR analysis using 3 plasmid-specific primer pairs (prapD, traA, and traL) and 3 chromosomal gene-specific primer pairs (yxaL, Rap6, and Rap12) of which the PCR product from yxaL primer pairs was detected in both donor and recipient and the other PCR products from Rap6 and Rap12 primer pairs were detected only in donor but not in recipient. Insertion of the full-length plasmid DNA was determined by PCR using 20 primer sets of the complete plasmid genome.

**Validation of differential gene expression**. Differential gene expression patterns in wild-type, plasmid-cured, and complemented cells were validated by reverse transcriptase-PCR and semi-quantitative PCR using the cDNA libraries prepared from the total RNA extracts at different times. In order to avoid contaminated or invalid samples, the quality of total RNA was checked by agarose gel electrophoresis to detect 16S and 23S ribosomal RNA bands and by a control PCR using primers directly on RNA and cDNA. In this study, a beta-propeller protein YxaL-coding gene was used as a constitutively expressed reference gene. The list of

gene-specific primer pairs used for the analysis of the relative gene expression normalized by the *yxaL* gene is given in Supplementary Table 4. Sigma factors for the transcription of plasmid-encoded *prapD* (BVH55_00015) orthogonal gene and the stringent Shine–Dalgarno sequence in the *B. subtilis* genome were searched by the previous methods[70–73].

**Analysis of proteins and antibiotics released into media**. Proteins and antibiotics (lipopeptides and polyketides) released from pBV71 plasmid-containing and cured cells into the culture media were analyzed at different incubation times (5, 8, 12, and 24 h) during culture in TSB with aeration (180 rpm) at 25 °C. A total volume of 20 mL of spent culture media was collected at each time point and filtered using 0.2 μm-membrane filters. Amicon Ultra Centrifugal filters with a cutoff of 3 kDa were used to concentrate 0.5 mL of this filtered culture fluid for analysis of protein bands by SDS PAGE with Coomassie blue staining, and western blotting for extracellular beta-propeller proteins, including the lateral root growth-promoting protein YxaL and 3-phytase Phy. A 9 mL portion of the filtered culture fluid was treated with 2 M trichloroacetic acid (TCA) for 1 h on ice in order to precipitate proteins, centrifuged at $16,000 \times g$ for 10 min, washed twice with ice-cold 80% acetone, and dissolved in 25 mM Tris/HCl buffer (pH 8.0) containing 55 mM dithiothreitol (DTT) for analysis of secretome by mass spectrometry. The remaining 10 mL of the filtered culture fluid was loaded to a Sep-Pak C18 column (bead volume 1 mL; Waters, Milford MA, USA) pre-washed with 10 mL acetonitrile (ACN) and equilibrated with 10% (v/v) ACN in 0.1% (v/v) trifluoroacetic acid (TFA). Then, the column was extensively washed with the same equilibration solution and 5 mL 50% ACN/0.1% TFA. The column-bound lipopeptides and polyketides were eluted with 5 mL 100% ACN. The vacuum dried samples were dissolved in 200 μL methanol, and insoluble particles were removed using 0.22 μm Costar Spin-X filters (Corning, Tewksbury, USA) before a 20 μL sample was injected into a Shimadzu 10AD HPLC system (Shimadzu, Kyoto, Japan) connected to a $4.6 \times 250$ mm Waters Sunfire C18 column. The analytical HPLC was operated with a 25 min linear gradient from 45% to 95% ACN in 0.05% TFA at a constant flow rate of 1 mL/min, and the column outlet was split for detection of UV absorbance at 230 nm by a Shimadzu SPD-10A UV-Vis detector and for the acquisition of a positive-mode electrospray mass spectrum in the range of $m/z$ 100 to 1800 by an API 2000 mass spectrometer (Applied Biosystems, Foster, USA). The base peak chromatogram by the HPLC-UV/mass spectrometry was used for the detection and quantification of antibiotics.

**SDS PAGE and western blot analysis**. The filtered supernatants were subjected to 10% SDS PAGE and Coomassie Brilliant Blue R250 staining for visualizing the released proteins that changed over time during the course of cell growth. Western blot analysis was also performed to determine the release of extracellular beta-propeller proteins, YxaL[12] and Phy[74], which have been known to promote plant growth. A rabbit polyclonal antibody for YxaL was produced and a mouse monoclonal antibody for PhyC (clone 3E3) was generated in this study with the C-terminal fragment (207–383 amino acid residues) of the full-length amino acid sequence of Phy (GenBank accession number AQS44456.1). The PhyC (207–383 a.a.)-coding DNA fragment in genomic DNA of strain GH1-13 was amplified using PCR primers (forward 5′-AGGAATTCATGCACCATCATCATCATCATCTGTCTGATCCTTAT CATTTTAC and reverse 5′-AGCTGCAGTTATTTTCCGCTTCTGTCAGTCA) with incorporation of the underlined EcoR1 and PstI restriction sites and a 6 × His tag in the N terminus of PhyC. The PCR amplicon was purified, digested, and cloned into the compatible restriction sites of plasmid pSD80, transformed, and induced in *E. coli* BL21 (DE3) according to the previously described method[75]. The IPTG-induced protein with a 6 × His tag was purified using Ni-NTA agarose (Qiagen, Hilden, Germany) and the buffer was exchanged to 1 × PBS before used as the antigen. The purified anti-YxaL and anti-PhyC IgG antibodies were used as primary antibodies at the concentration of 0.25 μg/mL in 5% skim milk-containing 1 × TBST buffer (20 mM Tris, 137 mM NaCl, and 0.1% Tween 20 in 1 L water adjusted to pH 7.6) for western blot. Appropriate secondary antibodies conjugated with horseradish peroxidase, Amersham Hybond P 0.45 PVDF blotting membrane, and ECL Prime Western Blotting detection reagents were obtained from Santa Cruz Biotechnology and GE Healthcare Life Science. Full images of the western blots and Coomassie blue-stained membranes for the filtered (concentrated) culture fluids of strain GH1-13 are shown in Supplementary Fig. 12.

**Analysis of RNA-seq data**. Total RNA from three replicate 1 mL cultures of wild-type pBV71 plasmid-containing and cured cells were isolated using RNeasy Mini kits according to the manufacturer's instructions (Qiagen, Valencia, USA). The three samples at each time point for each strain were pooled and analyzed as a single sample. RNA sequencing and alignment procedures were performed by ChunLab (Seoul, Korea). A Ribo-Zero rRNA removal kit (EpiCentre, Madison, USA) was used to deplete ribosomal RNA according to the manufacturer's instructions. Libraries for Illumina sequencing were prepared using a TruSeq Stranded mRNA sample prep kit (Illumina, San Diego, USA) following the manufacturer's protocol. RNA sequencing was performed on an Illumina HiSeq platform to generate single-ended 50-bp reads. Quality-filtered reads were aligned to the genome sequence of strain GH1-13 deposited at NCBI under the GenBank accession numbers CP019040.1 (chromosome) and CP019039.1 (plasmid). Relative

mRNA levels were measured in reads per kilobase of exon sequence per million (RPKM) mapped sequence reads[76], the relative log expression (RLE) mean[77], and the trimmed mean of the relative log expression (TMM) ratios[17]. The eggNOG (evolutionary genealogy of genes: non-supervised orthologous groups) database[78] was used to cluster the genes into functionally related groups, and the KEGG (Kyoto Encyclopedia of Genes and Genomes) database was used to analyze the metabolic pathways. The normalization procedures for differentially expressed gene (DEG) analysis were performed using ChunLab's CLRNASeq program. The RNA sequence read data of the two strains, named wild-type strain GH1-13 and plasmid-cured strain GH1-13cp, are available under NCBI BioProject PRJNA445855. To compare the gene expression patterns, raw mRNA counts were normalized to the trimmed mean of the relative log expression ratios (TMM), which resulted in a low coefficient of variation (CV) among the global normalization methods (Supplementary Fig. 8). Differential gene expression analysis of the $\log_2$-transformed TMM data was performed at a 99% of confidence level.

**Analysis of proteome**. Proteins in the filtered culture fluids (secretome) and whole-cell lysates (proteome) obtained at the different time points of cell growth were analyzed by tandem mass spectrometry. To obtain whole-cell lysates, harvested cells were washed twice with 25 mM Tris/HCl buffer (pH 8.0) at 4 °C, suspended in 3 volumes of denaturation buffer (8 M urea, 2% CHAPS, and 1 mM DTT in 25 mM Tris/HCl, pH 8.0), mixed with 1 volume of 0.5 mm glass beads (Biospec, Bartlesville, USA), and disrupted by five 50 s cycles in a Biospec bead beater with sample cooling on ice between cycles. After removing cell debris by centrifugation using a Beckman 70Ti rotor at 40,000 rpm and 4 °C for 1 h, protein concentrations in supernatants were determined using Bradford reagent and bovine serum albumin standard (BioRad, Hercules, USA). Protein samples, 10 μg of each (secretome and proteome) were treated with 10 mM DTT at 60 °C for 15 min to reduce disulfide bonds and cooled to room temperature before alkylation of cysteine with 50 mM iodoacetamide for 2 h in the dark. Proteins were precipitated by the addition of 2 M TCA for 1 h, centrifuged at $16,000 \times g$ for 15 min, and washed twice with 80% acetone at −20 °C. Air-dried proteins were treated with 0.5 μg of sequencing-grade trypsin (Promega, Madison, USA) in 25 mM ammonium bicarbonate buffer (pH 8.0) at 37 °C for 24 h and further digested for another 24 h with more addition of 0.5 μg trypsin. For proteome analysis of the whole-cell lysates, trypsinized samples were acidified with 0.5% TFA and loaded onto a pre-conditioned Waters MCX 1 cc column to fractionate peptides by stepwise elution with 5, 10, 15, 20, and 80% ACN in a 1.5% ammonia solution. All samples of secretome and proteome were dried in vacuo, dissolved in 0.1% TFA, and cleaned using Millipore ZipTip C18 tips. Peptide samples dissolved in 0.4% acetic acid were analyzed on a Thermo Velos Pro mass spectrometer equipped with a nanoflow liquid chromatography (nLC) systems and a reverse-phase Magic C18AQ column, 75 μm × 75 mm (Michrom Bioresources, Auburn, USA), operated at 0.3 μL/min. The chromatography condition was a linear gradient from 5% to 40% ACN in 0.1% formic acid solution for 50 min for secretome analysis and 90 min for proteome analysis, followed by a 20 min wash with 80% ACN and a 10 min re-equilibration to the initial condition. A full-scan survey was performed between $m/z$ 300 and 2000, followed by data-dependent MS2 scans of the 7 most intense ions from the survey scan. The acquired tandem mass spectrometry data were analyzed by Proteome Discoverer version 1.3 and Scaffold Proteome software version 4.4.5 using a protein database deduced from the chromosome and plasmid sequences of *B. velezensis* strain GH1-13 and the common protein contaminants in mass spectrometry. The SEQUEST search for tandem mass spectrum followed the options: average mass ($m/z$); maximum of 1 miscleavage site of trypsin digestion; precursor mass tolerance, 1.5 Da; fragment mass error, 1 Da; static modifications for carbamidomethylation of cysteine; protein probability >99%; and a false-discovery rate (FDR) < 0.01. Relative levels of proteins in each sample were calculated by the normalized spectral abundance factor (NSAF)[79] values. The mass spectrometry data of secretome and proteome have been deposited at the ProteomeXchange Consortium via the PRIDE[80] partner repository with the dataset identifiers PXD017305 and PXD008573, respectively.

**In-gel activity assays**. Non-denaturing blue native polyacrylamide gel electrophoresis was performed according to a previously described method[66]. Briefly, whole-cell extracts (20 μg protein load per lane) were obtained by cell disruption under ultrasonication in an ice bath followed by centrifugation. After electrophoresis on 10% polyacrylamide gels with cooling at 4 °C, enzyme activity was recovered by washing the gel with an appropriate buffer at room temperature for 10 min and transferred to the reaction solution composed of 10 μM phenazine methosulfate (PMS), 0.1 mM 3-(4,5-dimethylthiazol-2-yl)-2,5-diphenyltetrazolium bromide (MTT), 1 mM $NAD^+$ (or 1 mM $NADP^+$), and a fixed concentration of the substrate in the appropriate buffer: 10 mM glucose-6-phsophate for the NADP-dependent dehydrogenase (Zwf) in 100 mM sodium phosphate (pH 7.0), 10 mM 6-phosphogluconate for the NADP-dependent dehydrogenase (YqjI) in 100 mM sodium phosphate (pH 7.0), 10 mM glyceraldehyde-3-phosphate for the NAD-dependent dehydrogenase (GapB) in 100 mM sodium phosphate (pH 7.0), 20 mM sodium isocitrate for the NADP-dependent dehydrogenase (Icd) in 100 mM sodium phosphate (pH 7.0) containing 5 mM $MgCl_2$, 3 mM 2-oxogluconate for the NAD-dependent dehydrogenase (Odh) in 100 mM sodium phosphate (pH 7.2) containing 1 mM CoA, 84 mM sodium succinate for the NAD-dependent

dehydrogenase (Sdh) in 100 mM Tris/HCl (pH 7.5) containing 4.5 mM EDTA and 10 mM KCN, 20 mM L-malate for the NAD-dependent dehydrogenase (Mdh) in 100 mM Tris/HCl (pH 8.5), 10 mM 2,3-butanediol for the NAD-dependent dehydrogenase (BdhA) in 100 mM sodium phosphate (pH 7.0), and 10 mM L-threonine for the NAD-dependent dehydrogenase (Tdh) in 100 mM Tris/HCl (pH 8.0). While incubating the gels with 15 strokes per min at 25 °C in a dark chamber, in-gel enzyme activities were assayed by the formation of insoluble MTT formazan. The reaction was stopped by washing the gels with water, and the band intensity was calculated by determining the local average volume from the scan image with Molecular Dynamics ImageQuant software (version 5.2; GE Healthcare Life Science).

**Statistics and reproducibility**. Cell culture experiments were repeated at least three times independently, and results were reported as the means and standard deviations of the means. The data of repeated measurements are shown in Supplementary Data 3. A statistically significant difference between the experimental data of pBV71 plasmid-containing and cured strains was determined by two-tailed $t$-tests with $P$-values < 0.05. Using TMM normalized RNA-Seq data, analysis of variance ($F$-test) and chi-square test ($\chi^2$-test) with Yates' correction were performed to determine a significant change associated with the presence of pBV71. Post hoc comparisons of the pBV71 plasmid-associated genes were performed by Pearson's correlation analysis of the gene expression levels at different growth phases of the wild-type pBV71 plasmid-containing cells.

**Reporting summary**. Further information on research design is available in the Nature Research Reporting Summary linked to this article.

## Data availability

Microscopic data are available under EMBL-EBI BioStudies accession number S-BSST608. Transcriptome data are available under NCBI BioProject PRJNA445855. Secretome and proteome data have been deposited into PRIDE and are available via ProteomeXchange under the dataset identifiers PXD017305 and PXD008573, respectively.

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

## Acknowledgements

This work was supported by the National Research Foundation of Korea (NRF) grants funded by the Korea government (MSIT) (Grant No. NRF-2019R1F1A1062989 and NRF-2018R1A5A1025077) and by a cooperative research program for Agricultural Science & Technology Development, Rural Development Administration (Grant No. PJ01246706 and PJ01497402).

## Author contributions

The work was conceived by Y.H.K., C.W.L., and J.S, and designed by Y.H.K. Y.C., H.P., M.P.N., and L.V.H.T. performed the culture experiments, plasmid curing, complementation, conjugation, and RNA and protein analyses under the supervision of Y.H.K. J.K. performed the chemical analysis under the supervision of C.W.L. S.K. performed antagonistic tests on 6 plant pathogens under the supervision of J.S. J.S. supported the genome analysis. Y.H.K. drafted the manuscript, Y.H.K., C.W.L., and J.S. revised the manuscript and all authors discussed the results.

## Competing interests

The authors declare no competing interests.
