## [Peer Review File · Communications Biology]

Reviewers' comments:

Reviewer #2 (Remarks to the Author):

The study by Nguyen et al characterizes the physiological effects of a recently identified plasmid pV71. Though narrow in scope, this study reveals novel features of an interesting plasmid, was well executed, and therefore certainly merits publication. While I appreciate the findings, much of the results are observational, and I felt a few experimental data pieces and/or textual clarifications would greatly improve the rigor of the author's conclusions:

1. The plasmid curing process seems to be stressful for bacteria due to growth at heightened unfavorable temperatures. Ideally, whole genome sequencing would be presented to ensure the strain was indeed cured with no additional adaptations. However, as this is a potentially costly endeavor, the authors should at the minimum demonstrate reproducibility of key phenotypic results for some of the 18 strains that they cured (based on 6% of 300 strains). Perhaps that is what the authors did, but from the methods/text this wasn't clear.
2. Were any differences noted between WT and COM cells? Were the authors expecting to see a difference? I expected to see this, since the authors start the paper by highlighting that WT cells minus the plasmid lose the ability to produce IAA even though this function is not plasmid-encoded. They never follow up on why this might be, and all other results seem to suggest WT and COM show similar phenotypes. Clarification would go a long way at explaining the specific role of the plasmid itself, since some figures did not include COM comparisons, while others did, and any differences observed (if any) were not explained. Alternatively, the introduction should be restructured to avoid misleading the reader.
3. In line with #2, COM cells don't tell us much about plasmid-specific functions, as the host used was the PC strain, and therefore the two have already evolved together. It would have been more informative to use a host that had never carried pV71 previously in order to conclusively determine plasmid specificity. This was a key control that was missing in comparison to WT, PC, and COM.
4. The authors never sufficiently explain the cause for observed phenotype switching of plasmid cells.
5. There are some fairly strong conclusions that were not necessarily backed up by experimental data. The differences between WT and PC cells was striking with respect to metabolic gene regulation; I would have liked to see validation that these genes are altering some behavior. For example, the authors note changes in maltose metabolism, but cells were not even grown in maltose. What specifically is the effect of altered PTS? In general, I felt that there was an overall lacking in concrete links between observed data and overall conclusions. I believe just one or two demonstrative experiments with biochemical perturbations (e.g., changed growth media or otherwise) would sufficiently elaborate on this point.

Minor:

There are several grammatical and typographical errors throughout (e.g., line 185 "moderate correlated" should read "moderately correlated"; Line 75-76: "The most region of plasmid DNA encodes unknown" is difficult to understand. I imagine the authors intended to say "The largest region of plasmid DNA."; Line 106: "rapC in chromosome"; etc.).

Reviewer #3 (Remarks to the Author):

General comments.

In this paper the authors show the effects of presence or absence of plasmid pBV71 on the phenotype of strain GH1-13. They analyze physiological features, the morphology of the cells, as well as expression of genes and proteins, and protein secretion. The results show differences in the

morphology of the plasmid-cured derivative, in gene and protein expression and in some physiological responses. Some of the features had been analyzed in a previous paper (reference N° 11), but the analysis presented here is more comprehensive, although the conclusions are widely speculative. One important conclusion is that the plasmid affects chromosome-encoded functions, emphasizing the function of the genome as a whole.

My main concerns regard the definition of pBV71 as a conjugative plasmid, and the analysis of the plasmid-encoded genes. The paper does not show that the plasmid is able to perform conjugation, which would be decisive for this definition. The authors describe some genes which may be involved in conjugation, but their possible functions are not clearly explained, and no possible origin of transfer is identified. About the analysis of the plasmid-encoded genes, the authors mention that the plasmid has few sequences that show similarity in databases. I think it would be valuable to differentiate between hypothetical and conserved hypothetical genes. Even if the functions are unknown, it would be interesting to know with which organisms the sequences are shared. Taking 7 random protein sequences from plasmid sequence CP019039, I found that all of them had a very high similarity with hypothetical proteins from *B. subtilis*. Another inconvenience is that the protein nomenclature in the CP019039 database is different from the nomenclature used in the manuscript, making it difficult to identify and determine their localization in the plasmid.

Specific comments

1) The authors define pBV71 as a conjugative plasmid, because it has some genes related to conjugative transfer. As I understand, it has not been shown that the plasmid performs conjugative transfer, so that I think until its conjugative ability is demonstrated, it should only be considered as potentially conjugative, and the term "conjugative" eliminated from the title and the text.

Introduction

2) Lines 53 - 56: "The strain GH1-13 carries a unique plasmid (pBV71; 71,628 bp) that is poorly defined in existing databases due to low similarity and coverage. Curing of the plasmid in strain GH1-13 led to a decrease in the production of IAA11, but the plasmid does not carry this function."

- The authors should mention that reference 11 also analyzed how the absence of pBV71 affects colony morphology and biofilm production, reporting these findings: "that colonies of cured GH1-13cp were more mucous, smoother, and larger than intact GH1-13 when grown on agar plates. We examined the detailed the cell morphology of both cured and intact GH1-13 using transmission electron microscopy image (Fig. 3). Cell appeared to be of similar shape in both strains, although GH1-13cp had more acuminate edges based on the appearance of surrounding cell secretions. The intact GH1-13 strain appeared to have blunter edges (Fig. 3). The phenotypic differences observed between GH1-13 and GH1-13cp are likely associated with a different potential to produce an extracellular matrix through the secretion of exopolysaccharides responsible for biofilm formation (Branda et al., 2005). Biofilm formation was measured by using crystal violet according to the protocol by Yaryura et al. (2008). The result showed that GH1-13 strain has a better potential to form biofilm than GH1-13cp (Fig. 3)"

3) Lines 63 - 66: "The comparison of them showed that a native conjugative pBV71 plasmid played an important role in conjugation and biofilm formation of the host bacteria and their activation of envelope stress responses accompanying the extracellular polysaccharide biosynthesis and antibiotic production".

- The only result related to conjugation is the expression analysis of some conjugation-related genes. This is not sufficient evidence to say that the plasmid has an important role in conjugation.

4) Results

This section should be Results and Discussion, as there is no independent Discussion section.

5) Lines 70-72: "A 6 kb-region of the plasmid pBV71 in strain GH1-13 is similar to parts of Antarctic *B. safensis* strain U14-5 unnamed plasmid and 4 draft genome sequences of *Bacillus* species, as shown in Supplementary Fig. 1. These sequences share homology with genes involved in the conjugal transfer (Supplementary Table 1.)"

- The 6 kb-region contains 6 genes, which should be subject to a more profound analysis, in order to determine if they are involved in conjugative transfer. Three of the genes are similar to genes involved in the conjugation process (TrsD, TraC and TraL), while one is similar to cell wall hydrolases, and the two other genes are hypothetical. The possible functions of these genes should be analyzed.

In the sequence uploaded at NCBI (CP019039), the product of one coding sequence is defined as an aspartate phosphotransferase, and the other 93 coding sequences correspond to hypothetical proteins. I analyzed one of the sequences of the 6 kb-region. The protein encoded by WP_077721667.1, is much more similar to an hypothetical protein of *B. subtilis*, WP_069479640.1, with 100% cover and 89.5% identity, while the homologue from *B. safensis* is one among about ten other proteins from different organisms with cover about 90-95%, and identities between 60 and 70%, this is defined as the TraL.

6) Lines 73-75: "Transcript analysis of *B. velezensis* plasmid pBV71 showed that plasmid segregation and conjugal genes located in a replication initiation region of ~18 kb were highly expressed during early exponential phase of cells (Supplementary Fig. 2)"

- Please state which and where are the genes mentioned, and where the replication origin is located. Are the transfer genes clustered? Are they sufficient to provide the conjugation function? Is there a transfer origin?

7) Lines 75-78: "The most region of plasmid DNA encodes unknown or partial protein sequences without similarity in databases, as would be expected from multiple horizontal gene transfers mediated by conjugation through a type IV secretion system encoded by the *vir* and *tra* genes."

- I do not think this statement can be ascertained. It is more likely that multiple horizontal transfer events would lead to the plasmid carrying sequences with homology to diverse organisms. The annotation and similarity data for the plasmid sequences is needed to support this. Doing a random Blast search of 7 of the hypothetical proteins from the plasmid, all of them showed a very high conservation with hypothetical proteins from *B. subtilis*, indicating that these genes may be genera-specific.

- Also the redaction of the paragraph is not clear. Did the authors mean: "Most of the plasmid DNA encodes..."

8) Line 85: "This method determined 6% plasmid curing frequency on 300 selected colonies"

-This is a very high frequency for loss of a plasmid. This could be related to replication/segregation of the plasmid, or to other factors. The information regarding the function of the genes located on the plasmid could help to determine the reason for this high frequency of loss. Also, are there conditions where plasmid loss could be beneficial for the strain?

9) Lines 87 - 89: "examined (Fig. 1c-e and Supplementary Table 2), the WT parent and plasmid-cured (PC) variant showed similar patterns of growth and antibiotic susceptibility, except for the selenite and tellurite resistance of WT."

- The meaning of the result in Fig. 1e is not clear to me, and it seems contradictory to Fig. 1f. In Fig.1e, with 10 mg of sodium selenite the OD goes down to 0, in both the wild type and the PC strains, while in Fig. 1f you can see colonies growing in 500 mg.

- Are the genes allowing resistance to selenite and tellurite encoded in the plasmid or in the chromosome?

10) Lines 101-102: "In order to investigate a role of plasmid pBV71, as like a native conjugative plasmid of *B. subtilis* 13-19, expression levels of 14 aspartate phosphatase.."

- The meaning of this sentence is not clear to me

11) Line 108: " The plasmid rapD was expressed similarly as an ortholog of rapD in chromosome, and was probably..."

- Which of the genes corresponds to the chromosomal rapD?

12) Lines 113-114: "Simultaneous transcription of the plasmid rapD and chromosomal rap loci would activate such quorum-sensing modes that have been regulated by *B. subtilis* plasmid Rap proteins 16,18-20."

- To avoid speculation, why not check the expression of quorum sensing genes?

13) Lines 118-124: "(traA and traL) were co-expressed with phosphorelay genes including spo0F and degU encoding the response regulators that directly interact with many Rap proteins for dephosphorylation and jointly control complex colony development, and these were inversely correlated with the repression of phosphate metabolism regulator phoR22 (Fig. 5d and Supplementary Fig. 3). As in the high-density cells, this pattern appeared to correspond with the conjugation and biofilm formation in a quorum-sensing manner."

- The explanation is not accurate, traL expression seems much lower than that of any of the other genes mentioned, including phoR, whose expression seems higher in the PC strain. I think this is much too speculative.

14) Lines 157-160: " Fig. 7d shows that PC cells released high levels of Phy (two-tailed Z-test, $P < 0.05$), but had no significant difference of YxaL with WT and COM cells. This discrepancy indicated that plasmid pBV71 might not play only the role in the bacterial secretion system but also in the regulatory mechanism of the PGPR factors."

- The results of the section suggest that PC cells released lower levels of secreted proteins than the WT in late exponential phase. The basis of the last paragraph of the section mentioned above is not clear to me in this context.

15) Lines 194-195: "In the strain GH1-13, a thick capsule formation surrounded by the conjugative pili of plasmid pBV71 appears to be due.."

- It is not clear to me that this result is shown in the paper.

16) Fig. 3c "The head of an arrow shows conjugation between WT cells"

- I do not think this can be assured from the image, even if two cells are in contact through a pilus-like structure. First, there needs to be evidence of transfer of the plasmid.

Response Letter

Dear Editor and Reviewers,

I appreciate you very much for valuable comments on the manuscript entitled "Impact of a native conjugative plasmid on plant growth-promoting *Bacillus velezensis* strain GH1-13". I and co-authors revised the manuscript with more experiments according to the important points raised by the reviewers. I would like to submit the revised manuscript to Communications Biology, and explain all revisions that have been made in response to the reviewers' comments item by item, and highlight all changes in the manuscript file text. The manuscript has been edited by an English-speaking expert.

It has taken a longer time for the revision with more experiments than usual 3 months, due to changes in laboratory conditions and researchers between before and after COVID19. I hope that you understand that the order of the contributing authors' names is changed in the revised manuscript. Please provide me an author's agreement form for further filling.

I thank you for the opportunity to submit my work to Communications Biology.

Sincerely yours,

Yong-Hak Kim, PhD, Associate Professor,

Department of Microbiology, Daegu Catholic University School of Medicine,
Daegu 42472, Republic of Korea
ykim@cu.ac.kr

Response to comments of Reviewer #2:

The study by Nguyen et al characterizes the physiological effects of a recently identified plasmid pV71. Though narrow in scope, this study reveals novel features of an interesting plasmid, was well executed, and therefore certainly merits publication. While I appreciate the findings, much of the results are observational, and I felt a few experimental data pieces and/or textual clarifications would greatly improve the rigor of the author's conclusions:

1. The plasmid curing process seems to be stressful for bacteria due to growth at heightened unfavorable temperatures. Ideally, whole genome sequencing would be presented to ensure the strain was indeed cured with no additional adaptations. However, as this is a potentially costly endeavor, the authors should at the minimum demonstrate reproducibility of key phenotypic results for some of the 18 strains that they cured (based on 6% of 300 strains). Perhaps that is what the authors did, but from the methods/text this wasn't clear.

Author Response: It is a very important point to fully revise the manuscript. In the previous manuscript, the plasmid curing was estimated by the colony PCR using three gene-specific primer sets (*prapD*, *traA*, and *traL*) in the plasmid and a 16S rRNA gene-specific primer set. When PCR was performed using genomic DNA extracts, we found that 16 of the 18 colonies were positive for all three plasmid targets, and one of the remaining two mucoid colonies was

positive positive for *prapD* and *traA* but negative for *traL*, and the other one was negative for all three plasmid genes. To confirm the presence or absence of full-length plasmid DNA in each mucoid colony, PCR for the full-length plasmid genome was conducted using 20 primer pairs with head-to-tail and tail-to-head arrangements. Based on the results from more experiments, the plasmid curing rate was estimated at 0.3% of the total population. The results are described in a new section “Loss of pBV71 during conversion from non-mucoid to mucoid phenotypes”, with the revised data shown in Fig. 1b-c and Fig. 2d and f.

2. Were any differences noted between WT and COM cells? Were the authors expecting to see a difference? I expected to see this, since the authors start the paper by highlighting that WT cells minus the plasmid lose the ability to produce IAA even though this function is not plasmid-encoded. They never follow up on why this might be, and all other results seem to suggest WT and COM show similar phenotypes. Clarification would go a long way at explaining the specific role of the plasmid itself, since some figures did not include COM comparisons, while others did, and any differences observed (if any) were not explained. Alternatively, the introduction should be restructured to avoid misleading the reader.

Author Response: More experiments showed significant differences in antibiotic resistance and growth rates on lactose between non-mucoid and mucoid phenotypes of strain GH1-13 with or without the loss of the pBV71 plasmid. Moreover, the plasmid-cured (PC) cells fermented tagatose. When the plasmid was complemented into the PC cells, COM cells showed a non-mucoid phenotype similar to that of the WT cells which were resistant to bacitracin, gramicidin, selenite, and tellurite with reduced growth rates on lactose compared to mucoid phenotypes with or without pBV71 and with no detection of tagatose fermentation compared to that of the PC cells. This indicated that the presence of pBV71 in strain GH1-13 has a substantial effect on selection of the non-mucoid phenotype of catabolite repression and high antibiotic resistance, except for that ampicillin resistance varies between different variants of strain GH1-13 with or without pBV71. According to these findings, the Introduction and the Result sections of the revised manuscript were re-written and the revised data are shown in Fig. 5 and Supplementary Fig. 3, and Supplementary Tables 4 and 5.

3. In line with #2, COM cells don't tell us much about plasmid-specific functions, as the host used was the PC strain, and therefore the two have already evolved together. It would have been more informative to use a host that had never carried pV71 previously in order to conclusively determine plasmid specificity. This was a key control that was missing in comparison to WT, PC, and COM.

Author Response: Conjugation of the pBV71 plasmid into strain FZB42 with no plasmid was carried out to show that the conjugative pBV71 plasmid confers a selective advantage to host cells to increase the resistance to gramicidin and tellurite as similar to that of strain GH1-13. The result of the conjugation is described in the main text and the revised data are shown in Figs. 4 and 5, and Supplementary Fig. 3 and Supplementary Tables 4 and 5.

4. The authors never sufficiently explain the cause for observed phenotype switching of plasmid cells.

Author Response: The re-analysis of non-mucoid and mucoid colony isolates by PCR using genomic DNA extracts showed that a deficiency of the pBV71 plasmid in strain GH1-13 results in conversion from non-mucoid to mucoid phenotype and further loss (degradation) of the plasmid. However, this non-mucoid to mucoid conversion did not occur in another *B. velezensis* strain FZB42 and its transconjugant FZB42p containing pBV71. The strain GH1-13, but not strain FZB42, contains a new *rapF2-phrF2* pair that are highly expressed with the plasmid-encoded orphan *prapD* expression in WT and COM cells of strain GH1-13 in the early exponential phase. These results support that a unique RapF2-PhrF2 system might evolve incrementally into a QS system to manage the impact of pBV71 in strain GH1-13, but not in strain FZB42 and others. The revised results are shown in Figs. 1 to 5.

5. There are some fairly strong conclusions that were not necessarily backed up by experimental data. The differences between WT and PC cells was striking with respect to metabolic gene regulation; I would have liked to see validation that these genes are altering some behavior. For example, the authors note changes in maltose metabolism, but cells were not even grown in maltose. What specifically is the effect of altered PTS? In general, I felt that there was an overall lacking in concrete links between observed data and overall conclusions. I believe just one or two demonstrative experiments with biochemical perturbations (e.g., changed growth media or otherwise) would sufficiently elaborate on this point.

Author Response: Growth and fermentation characteristics of wild-type and mutant strains GH1-13 and FZB42 showed that strain GH1-13, but not strain FZB42, could increase growth rates on lactose upon the conversion to mucoid phenotype independent of the presence of pBV71, and the plasmid loss caused an increased fermentation of tagatose. Complementation of the plasmid produced a non-mucoid phenotype similar to that of wild-type cells with reduced growth rates on lactose and with no tagatose fermentation detected by API 50 CH medium. These results, along with multi-omics data, suggest the role of the pBV71 plasmid in the regulation of non-mucoid phenotype of strain GH1-13, but not that of strain FZB42, associated with catabolite repression of lactose and tagatose, although these genes have not yet been functionally defined in *B. velezensis*. These results are shown in Fig. 5, Supplementary Fig. 3, and Supplementary Table 5.

Minor:

There are several grammatical and typographical errors throughout (e.g., line 185 “moderate correlated” should read “moderately correlated”; Line 75-76: “The most region of plasmid DNA encodes unknown” is difficult to understand. I imagine the authors intended to say “The largest region of plasmid DNA.”; Line 106: “rapC in chromosome”; etc.).

Author Response: The English editing was made by a professional English native speaker.

I thank you very much for valuable comments on my paper.

Response to comments of Reviewer #3:

General comments.

In this paper the authors show the effects of presence or absence of plasmid pBV71 on the phenotype of strain GH1-13. They analyze physiological features, the morphology of the cells, as well as expression of genes and proteins, and protein secretion. The results show differences in the morphology of the plasmid-cured derivative, in gene and protein expression and in some physiological responses. Some of the features had been analyzed in a previous paper (reference N° 11), but the analysis presented here is more comprehensive, although the conclusions are widely speculative. One important conclusion is that the plasmid affects chromosome-encoded functions, emphasizing the function of the genome as a whole.

My main concerns regard the definition of pBV71 as a conjugative plasmid, and the analysis of the plasmid-encoded genes. The paper does not show that the plasmid is able to perform conjugation, which would be decisive for this definition. The authors describe some genes which may be involved in conjugation, but their possible functions are not clearly explained, and no possible origin of transfer is identified. About the analysis of the plasmid-encoded genes, the authors mention that the plasmid has few sequences that show similarity in databases. I think it would be valuable to differentiate between hypothetical and conserved hypothetical genes. Even if the functions are unknown, it would be interesting to know with which organisms the sequences are shared. Taking 7 random protein sequences from plasmid sequence CP019039, I found that all of them had a very high similarity with hypothetical proteins from *B. subtilis*. Another inconvenience is that the protein nomenclature in the CP019039 database is different from the nomenclature used in the manuscript, making it difficult to identify and determine their localization in the plasmid.

Author Response: We thank the reviewer for valuable comments on the revision of this paper with more experiments for the conjugative pBV71 plasmid and the function in the host. We could revise the manuscript carefully point-by-point according to the specific comments. The revised BLAST analysis results for protein-coding DNA sequences and their locations in the plasmid of strain GH1-13, named pBV71, are shown in Supplementary Table 1.

Specific comments

1) The authors define pBV71 as a conjugative plasmid, because it has some genes related to conjugative transfer. As I understand, it has not been shown that the plasmid performs conjugative transfer, so that I think until its conjugative ability is demonstrated, it should only be considered as potentially conjugative, and the term "conjugative" eliminated from the title and the text.

Author Response: The native pBV71 plasmid was conjugated into strain FZB42 with no plasmid. The result of the conjugation is shown in Fig. 4 in the main text.

Introduction

2) Lines 53 - 56: "The strain GH1-13 carries a unique plasmid (pBV71; 71,628 bp) that is poorly defined in existing databases due to low similarity and coverage. Curing of the plasmid in strain GH1-13 led to a decrease in the production of IAA11, but the plasmid does not carry this function."

- The authors should mention that reference 11 also analyzed how the absence of pBV71 affects colony morphology and biofilm production, reporting these findings: "that colonies of cured GH1-13cp were more mucous, smoother, and larger than intact GH1-13 when grown on agar plates. We examined the detailed the cell morphology of both cured and intact GH1-13 using transmission electron microscopy image (Fig. 3). Cell appeared to be of similar shape in both strains, although GH1-13cp had more acuminate edges based on the appearance of surrounding cell secretions. The intact GH1-13 strain appeared to have blunter edges (Fig. 3). The phenotypic differences observed between GH1-13 and GH1-13cp are likely associated with a different potential to produce an extracellular matrix through the secretion of exopolysaccharides responsible for biofilm formation (Branda et al., 2005). Biofilm formation was measured by using crystal violet according to the protocol by Yaryura et al. (2008). The result showed that GH1-13 strain has a better potential to form biofilm than GH1-13cp (Fig. 3)"

Author Response: The revised experiments showed that the non-mucoid phenotype of wild-type strain GH1-13 was often converted to a mucoid phenotype with or without the loss of the plasmid during normal growth conditions in rich media. The resulting mucoid phenotypes were similar to that of the heat-cured strain GH1-13cp in the previous study. According to the PCR results obtained using 20 primer sets for the full-length plasmid DNA, it was shown that one of the 18 mucoids among 300 randomly selected colonies had lost almost all of the plasmid genome, which gave an estimated curing rate of 0.3% of the total population. Although the number of mucoids is lower than the number of non-mucoids, it may be likely that a deficiency of pBV71 in a sub-population of host cells will present a mucoid phenotype. To explain a high potential of biofilm formation by non-mucoid phenotypes compared to non-mucoid phenotypes of strain GH1-13, we performed more experiments with conjugation of the native pBV71 from a mucoid variant (MU3) into a stable non-mucoid phenotype of strain FZB42 and gene expression analysis of quorum-sensing systems in the two strains and their variants with or without plasmid pBV71. The revised data and results are described in the main text and Figs 1 to 5.

3) Lines 63 - 66: "The comparison of them showed that a native conjugative pBV71 plasmid played an important role in conjugation and biofilm formation of the host bacteria and their activation of envelope stress responses accompanying the extracellular polysaccharide biosynthesis and antibiotic production".

- The only result related to conjugation is the expression analysis of some conjugation-related genes. This is not sufficient evidence to say that the plasmid has an important role in conjugation.

Author Response: The native pBV71 plasmid was conjugated into *B. velezensis* strain FZB42 with no plasmid.

4) Results

This section should be Results and Discussion, as there is no independent Discussion section.

Author Response: The section title of Results was changed to Results and Discussion.

5) Lines 70-72: "A 6 kb-region of the plasmid pBV71 in strain GH1-13 is similar to parts of Antarctic *B. safensis* strain U14-5 unnamed plasmid and 4 draft genome sequences of *Bacillus* species, as shown in Supplementary Fig. 1. These sequences share homology with genes involved in the conjugal transfer (Supplementary Table 1.)"

- The 6 kb-region contains 6 genes, which should be subject to a more profound analysis, in order to determine if they are involved in conjugative transfer. Three of the genes are similar to genes involved in the conjugation process (TrsD, TraC and TraL), while one is similar to cell wall hydrolases, and the two other genes are hypothetical. The possible functions of these genes should be analyzed.

In the sequence uploaded at NCBI (CP019039), the product of one coding sequence is defined as an aspartate phosphotransferase, and the other 93 coding sequences correspond to hypothetical proteins. I analyzed one of the sequences of the 6 kb-region. The protein encoded by WP_077721667.1, is much more similar to an hypothetical protein of *B. subtilis*, WP_069479640.1, with 100% cover and 89.5% identity, while the homologue from *B. safensis* is one among about ten other proteins from different organisms with cover about 90-95%, and identities between 60 and 70%, this is defined as the TraL.

Author Response: Conjugation of the native pBV71 plasmid into a closely related strain FZB42 with no plasmid was shown in the revised manuscript, although it is difficult to specify the functions of the annotated genes and hypothetical genes in a conserved 6 kb-region of the plasmid. In this study, we focused on the effects of the pBV71 plasmid on host cells conferring a selective advantage by increasing resistance to antibiotics and toxic metal ions with physiological and morphological changes. The expression of plasmid-encoded aspartate phosphotransferase pRapD accorded a new quorum-sensing system of RapF2-PhrF2 present in the chromosome of strain GH1-13, but not in strain FZB42, which showed that there is intimate crosstalk between their expression, cell morphogenesis, carbon catabolite repression, antibiotic production, biofilm and spore formation in the non-mucoid phenotype of strain GH1-13. The revised data are shown in Figs. 4 and 5, and Supplementary Tables 4 and 5.

6) Lines 73-75: "Transcript analysis of *B. velezensis* plasmid pBV71 showed that plasmid segregation and conjugal genes located in a replication initiation region of ~18 kb were highly

expressed during early exponential phase of cells (Supplementary Fig. 2)"

- Please state which and where are the genes mentioned, and where the replication origin is located. Are the transfer genes clustered? Are they sufficient to provide the conjugation function? Is there a transfer origin?

Author Response: The sentence was corrected in the revised paper. "We noticed that several ORFs encoding putative conjugal proteins in a ~18-kb region between the plasmid segregation protein ParM gene and a type IV secretion system protein VirD4 (TrsK) gene were highly expressed during the early-exponential phase (5 h culture) of the host cells. However, most hypothetical genes scattered on the plasmid make it difficult to specify their role in the conjugation process."

7) Lines 75-78: "The most region of plasmid DNA encodes unknown or partial protein sequences without similarity in databases, as would be expected from multiple horizontal gene transfers mediated by conjugation through a type IV secretion system encoded by the *vir* and *tra* genes."

- I do not think this statement can be ascertained. It is more likely that multiple horizontal transfer events would lead to the plasmid carrying sequences with homology to diverse organisms. The annotation and similarity data for the plasmid sequences is needed to support this. Doing a random Blast search of 7 of the hypothetical proteins from the plasmid, all of them showed a very high conservation with hypothetical proteins from *B. subtilis*, indicating that these genes may be genera-specific.

- Also the redaction of the paragraph is not clear. Did the authors mean: "Most of the plasmid DNA encodes..."

Author Response: The paragraph was corrected with removing the ascertain statement and correcting the typographical error in the revised manuscript. The BLAST results of protein-coding sequences in the plasmid genome is shown in Supplementary Table 1.

8) Line 85: "This method determined 6% plasmid curing frequency on 300 selected colonies"

-This is a very high frequency for loss of a plasmid. This could be related to replication/segregation of the plasmid, or to other factors. The information regarding the function of the genes located on the plasmid could help to determine the reason for this high frequency of loss. Also, are there conditions where plasmid loss could be beneficial for the strain?

Author Response: It is a very important point enabling us to fully revise the manuscript. In the previous manuscript, the plasmid curing was estimated by the colony PCR using three gene-specific primer sets (*prapD*, *traA*, and *traL*) in the plasmid and a 16S rRNA gene-specific primer set. When PCR was performed using genomic DNA extracts, we found that 16 of the 18 colonies were positive for all three plasmid targets, and one of the remaining two mucoid colonies was positive positive for *prapD* and *traA* but negative for *traL*, and the other one was negative for all three plasmid genes. To confirm the presence or absence of full-length plasmid

DNA in each mucoid colony, PCR for the full-length plasmid genome was conducted using 20 primer pairs with head-to-tail and tail-to-head arrangements. Based on the results from more experiments, the plasmid curing rate was estimated at 0.3% of the total population. The results are described in a new section "Loss of pBV71 during conversion from non-mucoid to mucoid phenotypes", with the revised data shown in Fig. 1b-c and Fig. 2d and f.

9) Lines 87 - 89: "examined (Fig. 1c-e and Supplementary Table 2), the WT parent and plasmid-cured (PC) variant showed similar patterns of growth and antibiotic susceptibility, except for the selenite and tellurite resistance of WT."

- The meaning of the result in Fig. 1e is not clear to me, and it seems contradictory to Fig. 1f. In Fig. 1e, with 10 mg of sodium selenite the OD goes down to 0, in both the wild type and the PC strains, while in Fig. 1f you can see colonies growing in 500 mg.

- Are the genes allowing resistance to selenite and tellurite encoded in the plasmid or in the chromosome?

Author Response: Antibiotic susceptibilities of strains GH1-13 and FZB42 as well as their variants with or without the pBV71 plasmid were more extensively analyzed by the minimum inhibitory concentration (MIC) method and the results are shown in Supplementary Table 4. The revised experiments showed that strain GH1-13 became sensitive to bacitracin and gramicidin as well as selenite and tellurite upon the conversion to mucoid phenotypes with or without the loss of the plasmid. The antibiotic resistance was rescued in complemented cells with the non-mucoid phenotype similar to that of the wild-type cells. Conjugation of the plasmid into strain FZB42 also increased resistance to gramicidin and tellurite similar as that did the non-mucoid phenotype of strain GH1-13 with the plasmid.

The mistake of writing a wrong concentration of tellurite and inconsistent concentration units in the previous figure 1e are converted to 500 ug/mL selenite and 20 ug/mL tellurite. The misreading in the MIC test in the previous figures 1e was removed because measurement of the optical density of cells at 600 nm has a large error with the reduction of low concentration selenite to red pigment with absorbance at 600 nm. The MIC values of selenite for the non-mucoid (5 mg/mL) and mucoid (2.5 mg/ml) phenotypes of strain GH1-13 with or without the plasmid pBV71 are directly shown in a photograph (Fig. 1f) taken after 24 h incubation of the cells in a 96-well microplate containing serial 2-fold dilutions of sodium selenite from 5 to 0.08 mg/mL in TSB medium.

Antibiotic sensitivities of strains GH1-13 and FZB42 with similar genetic backgrounds, but for the plasmid, appeared to vary depending on the phenotype conversion or plasmid presence which may be related to the expression of the phage-shock proteins LiaIH and the synthesis of lipopeptide surfactin in the cell envelope and extracellular layers, which have suggested a direct role in the resistance to bacitracin and gramicidin in previous studies (Rautenbach et al. 2012. *Microbiology*. **158**, 3072-3082; Yu et al. 2018. *Environ. Sci. Technol.* **52**, 10400-10407), and alterations in envelope sulfhydryl sites and thioredoxin disulfide reductase TrxR that control the adsorption and metabolism of selenite and tellurite, resulting in metal forms in the cells (Yasir et al. 2020. *FEMS Microbiol. Ecol.* **96**, fiaa126; Van der Heiden et al. 2013. *Appl. Environ. Microbiol.* **79**, 3511-3515). Therefore, in our study, the effects of plasmid pBV71 on host cells were extensively evaluated by transmission electron microscopy images of tellurite-treated cells,

antagonistic effects against *Fusarium solani*, quantitative HPLC-mass spectrometry analysis of macrolactin and surfactin antibiotics, and multi-omics approaches.

10) Lines 101-102: "In order to investigate a role of plasmid pBV71, as like a native conjugative plasmid of *B. subtilis*13-19, expression levels of 14 aspartate phosphatase.."

- The meaning of this sentence is not clear to me

Author Response: This Result section was fully revised based on the revised experiment results: "To explore the potential role of pBV71 in the linkage between phenotype change and catabolite control in the host, ...

11) Line 108: " The plasmid rapD was expressed similarly as an ortholog of rapD in chromosome, and was probably..."

- Which of the genes corresponds to the chromosomal rapD?

Author Response: This ascertain sentence was removed in the revised manuscript.

12) Lines 113-114: "Simultaneous transcription of the plasmid rapD and chromosomal rap loci would activate such quorum-sensing modes that have been regulated by *B. subtilis* plasmid Rap proteins16,18-20."

- To avoid speculation, why not check the expression of quorum sensing genes?

Author Response: Semi-quantitative RT-PCR results for quorum-sensing (QS) systems (*comQXAP* and *rap-phr* pairs) in different variants of strains GH1-13 and FZB42 showed positive relationships between the expression of *prapD* and chromosomal *rapF2-phrF2* genes in the early exponential phase of wild-type and complemented cells of strain GH1-13 (Fig. 5c). This supports that a unique RapF2-PhrF2 system may evolve incrementally into a QS system to manage the impact of pBV71 in strain GH1-13, but not in strain FZB42 and others.

13) Lines 118-124: "(traA and traL) were co-expressed with phosphorelay genes including spo0F and degU encoding the response regulators that directly interact with many Rap proteins for dephosphorylation and jointly control complex colony development, and these were inversely correlated with the repression of phosphate metabolism regulator phoR22 (Fig. 5d and Supplementary Fig. 3). As in the high-density cells, this pattern appeared to correspond with the conjugation and biofilm formation in a quorum-sensing manner."

- The explanation is not accurate, traL expression seems much lower than that of any of the other genes mentioned, including phoR, whose expression seems higher in the PC strain. I think this is much to speculative.

Author Response: This Result section was fully re-written based on the analysis of plasmid-encoded pRapD and chromosomal quorum-sensing gene expression. The wrong sentence “inversely correlated with the repression of phosphate metabolism regulator *phoR*” is converted to “reversely correlate with the expression of phosphate metabolism regulator”, as pointed by the reviewer. Schematic illustration of the revised model, showing the impact of pBV71 on conjugation and biofilm formation of host cells in a quorum-sensing manner, is shown in Fig. 6d.

14) Lines 157-160: " Fig. 7d shows that PC cells released high levels of Phy (two-tailed Z-test, $P < 0.05$), but had no significant difference of YxaL with WT and COM cells. This discrepancy indicated that plasmid pBV71 might not play only the role in the bacterial secretion system but also in the regulatory mechanism of the PGPR factors."

- The results of the section suggest that PC cells released lower levels of secreted proteins than the WT in late exponential phase. The basis of the last paragraph of the section mentioned above is not clear to me in this context.

Author Response: The unclear section was re-written and a reference was added. “PC cells increased the secretion of Phy compared to that did the WT and COM cells in all four phases (two-tailed Z-test, $P < 0.05$), but there were no significant differences in the secretion pattern of YxaL among the three strains. These results are consistent with the expression patterns of constitutive *yxaL* (Kim et al. 2019. *PLoS One*. **14**, e0207968) and phosphate starvation-inducible *phyC* under the control of PhoPR (Makarewicz et al. 2006. *J. Bacteriol.* **188**, 6953-6965). Because Makarewicz et al. (2006) suggested that the *phyC* expression is up- or down-regulated by altering the level of phosphorylated PhoP~P through the regulation of PhoP-PhoR two-component system in *B. velezensis* FZB42, increased secretion of PhyC is strongly related to the up-regulation of *phoR* expression in PC cells compared to that in the WT and COM cells in early exponential phase.”

The last sentence was also re-written. “This suggests that the conjugative plasmid expression affects the expression pattern of some PGPR traits as well as the physiological and morphological traits of the host.”

15) Lines 194-195: "In the strain GH1-13, a thick capsule formation surrounded by the conjugative pili of plasmid pBV71 appears to be due.."

- It is not clear to me that this result is shown in the paper.

Author Response: The unclear sentence was removed in the revised manuscript.

16) Fig. 3c "The head of an arrow shows conjugation between WT cells"

- I do not think this can be assured from the image, even if two cells are in contact through a

pilus-like structure. First, there needs to be evidence of transfer of the plasmid.

Author Response: This sentence was corrected in the legend of Fig. 3c. “The head of the arrow shows a pilus-like structure that connects cells across the space.” During the conjugation of the native pBV71 plasmid from a mucoid variant (MU3) of strain GH1-13 as the donor to the recipient strain FZB42, the co-culture showed longer pilus-like structures that connected cells across the loose extracellular matrix, possibly as a conjugation bridge for the transfer of plasmid DNA (Fig. 3d).

I thank you very much for valuable comments on my paper.

REVIEWERS' COMMENTS:

Reviewer #2 (Remarks to the Author):

The authors have satisfactorily addressed all of my concerns and I find the manuscript much improved.